# BAH Dataset for Ambivalence/Hesitancy Recognition in Videos for Digital Behavioural Change

**Manuela González-González**[3,4], **Soufiane Belharbi**[1], **Muhammad Osama Zeeshan**[1],
**Masoumeh Sharafi**[1], **Muhammad Haseeb Aslam**[1], **Marco Pedersoli**[1],
**Alessandro Lameiras Koerich**[2], **Simon L Bacon**[3,4] & **Eric Granger**[1]

[1]LIVIA, Dept. of Systems Engineering, ETS Montreal, Canada
[2]LIVIA, Dept. of Software and IT Engineering, ETS Montreal, Canada
[3]Dept. of Health, Kinesiology, & Applied Physiology, Concordia University, Montreal, Canada
[4]Montreal Behavioural Medicine Centre, CIUSSS Nord-de-l'Ile-de-Montréal, Canada

{soufiane.belharbi, marco.pedersoli, alessandro.koerich, eric.granger}@etsmtl.ca
{muhammad-osama.zeeshan.1,masoumeh.sharafi.1,muhammad-haseeb.aslam.1}@ens.etsmtl.ca
manuela.gonzalez@mail.concordia.ca, simon.bacon@concordia.ca

## Abstract

Ambivalence and hesitancy (A/H), closely related constructs, are the primary reasons why individuals delay, avoid, or abandon health behaviour changes. They are subtle and conflicting emotions that sets a person in a state between positive and negative orientations, or between acceptance and refusal to do something. They manifest as a discord in affect between multiple modalities or within a modality, such as facial and vocal expressions, and body language. Although experts can be trained to recognize A/H as done for in-person interactions, integrating them into digital health interventions is costly and less effective. Automatic A/H recognition is therefore critical for the personalization and cost-effectiveness of digital behaviour change interventions. However, no datasets currently exist for the design of machine learning models to recognize A/H. This paper introduces the Behavioural Ambivalence/Hesitancy (BAH) dataset collected for multimodal recognition of A/H in videos. It contains 1,427 videos with a total duration of 10.60 hours, captured from 300 participants across Canada, answering predefined questions to elicit A/H. It is intended to mirror real-world digital behaviour change interventions delivered online. BAH is annotated by three experts to provide timestamps that indicate where A/H occurs, and frame- and video-level annotations with A/H cues. Video transcripts, cropped and aligned faces, and participant metadata are also provided. Since A and H manifest similarly in practice, we provide a binary annotation indicating the presence or absence of A/H. Additionally, this paper includes benchmarking results using baseline models on BAH for frame- and video-level recognition, zero-shot prediction, and personalization with source-free domain adaptation methods. The limited performance highlights the need for adapted multimodal and spatio-temporal models for A/H recognition. Results obtained with specialized fusion methods are shown to assess the presence of conflicts between modalities, additionally temporal modelling for within-modality conflicts are essential for more discriminant A/H recognition. The data, code, and pretrained weights are publicly available: github.com/LIVIAETS/bah-dataset.

## 1 Introduction

Emotion recognition plays a growing role in a range of health-related domains (Siddiqi et al., 2024), including disease prevention (Jin, 2024), diagnosis (Jiang et al., 2024; Maki et al., 2013), treat-

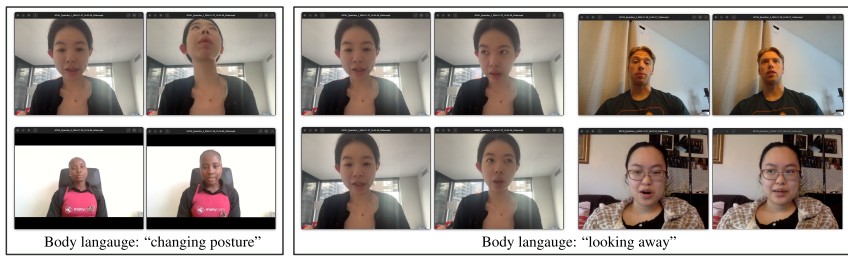

Figure 1: Examples of body language cues used by annotators to identify the occurrence of A/H: "looking away," and "changing posture."

ment monitoring (Dhuheir et al., 2021; Pepa et al., 2021; Suraj et al., 2022), and digital health promotion (Arabian et al., 2023; Subramanian et al., 2022), by supporting adaptive and responsive interventions (Liu et al., 2024b; Sinha et al., 2020). Emotion recognition technologies can support behaviour change interventions (Guo et al., 2024) by identifying affective states relevant to motivation, adherence, and engagement. Health-related behaviour change focuses on strategies to support individuals in adopting and maintaining healthy behaviours to prevent or manage diseases, reduce early mortality, and improve mental health and well-being (Davidson & Scholz, 2020). Achieving and maintaining long-term behaviour change is a complex process. It often includes overcoming ambivalence and hesitancy (A/H) (McDonald et al., 2002; Michie et al., 2013a;b; Voisard et al., 2024), closely related constructs, which are the primary reasons for individuals to delay, avoid, or abandon health behaviour changes (Conner & Armitage, 2008; Conner & Sparks, 2002; Manuel & Moyers, 2016; Miller & Rose, 2015; Van Gent et al., 2024; Williams, 2024). A/H is a subtle and conflicting emotion manifested by a discord in affect between multiple modalities or within a modality, such as facial and vocal expressions, and body language. This conflict sets a person in a state between positive and negative orientations, or between acceptance and refusal to do something, which often constitutes a barrier to initiating behaviour change and a trigger for discontinuing interventions or change efforts. Healthcare providers (e.g., clinicians, therapists) often identify A/H through a combination of speech and non-verbal cues (e.g., facial expressions and tone) (Heisel & Mongrain, 2004; Labbé et al., 2022; Miller & Rose, 2015) during in-person interactions. However, integrating them into digital interventions and e-health is costly and less effective. Therefore, designing robust automated methods for A/H recognition can provide a cost-effective alternative that can adapt to individual users and operate seamlessly in real-time and resource-limited environments.

Recent research on machine learning (ML) in emotion recognition focuses mainly on seven basic discrete emotions in images or videos, e.g, 'Happy', 'Sad', and 'Surprised' (Belharbi et al., 2024a; Liu et al., 2024a; Xue et al., 2022). Other models in the literature predict ordinal levels, including levels linked to health states, e.g., pain, depression, and stress estimation (Aslam et al., 2024; Chaptoukaev et al., 2023; Zeeshan et al., 2024; Nasimzada et al., 2024). Other affects are continuous predictions such as valence-arousal (Dong et al., 2024; Praveen & Alam, 2024a; Praveen et al., 2023; 2021). However, real-world scenarios present more complex cases of emotions. Recently, there has been an increased interest in designing robust affect models for compound emotions, a case where a mixture of basic emotions is manifested (Kollias, 2023; Richet et al., 2024). In particular, compound emotions commonly occur in daily interactions. Recognizing affective states linked to compound emotions and health states are however, more difficult to discern as they are subtle, ambiguous, and resemble basic emotions. A/H recognition is related to such tasks where intention and attitudes are conflicting or in an intermediate (in-between) state, between willingness and resistance (MacDonald, 2015), or positive and negative affect (Armitage & Conner, 2000). This can manifest in how individuals express themselves and can be recognized (Hayashi et al., 2023) in their facial expression, tone, verbal expressions, and body language (Figure 1). As a result, A/H is inherently multimodal, arising from subtle interactions among multiple cues both across modalities and within each modality. Unfortunately, such discord is extremely difficult to detect – a task that requires human training. This is a tedious and expensive procedure, leading to ineffective and less scalable digital interventions under limited resources. Integrating automated and reliable tools for A/H recognition can have a major impact on improving digital health interventions. Although A/H is a common topic in behavioural science (Conner & Armitage, 2008; Hohman et al., 2016; Manuel & Moyers, 2016), it remains unexplored in the ML community, and as such, in the design of eHealth

components. A possible reason is the lack of the necessary and specialized data for training and evaluating ML models.

To address this limitation, this paper introduces a first Behavioural Ambivalence/Hesitancy (BAH) dataset collected for subject-based multimodal recognition of A/H in videos. Through a collaboration between behavioural science and ML teams, we have collected a large video dataset, BAH, from 9 provinces in Canada. A data capture protocol was set in place to recruit diverse participants, including the development of a web platform for video capturing, a dedicated storage server, and a specific annotation protocol. Our behavioural science team designed seven questions to elicit responses regarding behaviours and to identify possible instances where participants displayed A/H. On our web-platform, participants were presented with these questions and asked to record themselves while answering, using the camera and microphone on their device. Participants were guided on the platform by an avatar throughout the entire data capture session. Our dataset was developed with real-world digital health applications in mind, particularly for use in personalized behaviour change interventions where users interact with an avatar that prompts them with predefined questions. To mirror this setting, participants in our study responded to structured but genuine questions about behaviours they engage in or avoid, based on their actual experiences. This setup encourages authentic, spontaneous expression of ambivalence and hesitancy (A/H), while still maintaining consistency across participants. The BAH dataset was intentionally designed to maximize ecological validity. Participants had control over their environment and delivery, which contributed to naturalistic responses that reflect how people express A/H in real life.

The BAH dataset is composed of videos from 300 participants. This amounts to a total of 1,427 videos ($\sim$ 10.60 hours) where 778 videos contain A/H ($\sim$ 1.79 hours). It has 916,618 total frames, where 156,255 contain A/H. Three of our behavioural experts annotated the data at the video- and frame-level to assess when A/H occurs. This also provides timestamps indicating when A/H starts and ends, allowing temporal localization. Because ambivalence and hesitancy manifest similarly in practice, the annotation is framed under a binary form (A/H vs. non-A/H). The video cues used by the annotators are also reported, such as facial expressions, body language, audio and language, in addition to highlighting where there is inconsistency between the modalities. The BAH dataset is made public, and it is provided with the raw videos with audio, cropped and aligned faces, detailed annotation/cues at the video- and frame-level, A/H timestamps, audio transcript/timestamps/language, and participant metadata such as age, ethnicity, etc. Part of BAH has already been used in a public challenge at the 2025 (Hallmen et al., 2025; Kollias et al., 2025; Savchenko, 2025) and 2026 'Workshop and Competition on Affective & Behavior Analysis in-the-wild (ABAW)' (8th and 10th ABAW).

**Our main contributions are summarized as follows.** **(1)** A novel video dataset named BAH is proposed for automated participant-based and multimodal recognition of A/H in videos that can be used to design ML models for digital health intervention systems. To mirror real-world behaviour change interventions, 300 participants in our study recorded themselves while responding to structured but genuine questions about behaviours that elicit A/H using our online platform. Behavioural experts labelled the data at the video- and frame-levels to indicate the presence/absence of A/H (since they are manifested similarly). BAH can be used to develop and evaluate standard and personalized ML models for classification and localization tasks, and build insights about A/H for digital behaviour change interventions. **(2)** Preliminary benchmarking results using baseline models on BAH for frame- and video-based emotion recognition. Results allowed exploring the impact of key factors, including the impact of using temporal context, multimodal information, and feature fusion. They also showed the need for specialized modalities fusion, and temporal modelling for better performance in the new task of A/H recognition. Baseline results are also shown for other tasks – zero-shot prediction and personalization (source-free domain adaptation) through subject-based domain adaptation. Our code and dataset are publicly available.

## 2 THE BAH DATASET

### 2.1 DATASET COLLECTION AND ANNOTATION

**Capture**. The BAH dataset contains Q&A videos. It is constructed by collecting samples from participants over the age of 18 across Canada. The data collection and annotation process is presented in Figure 2. To proceed with the data collection, we developed an "Automatic Expression Recogni-

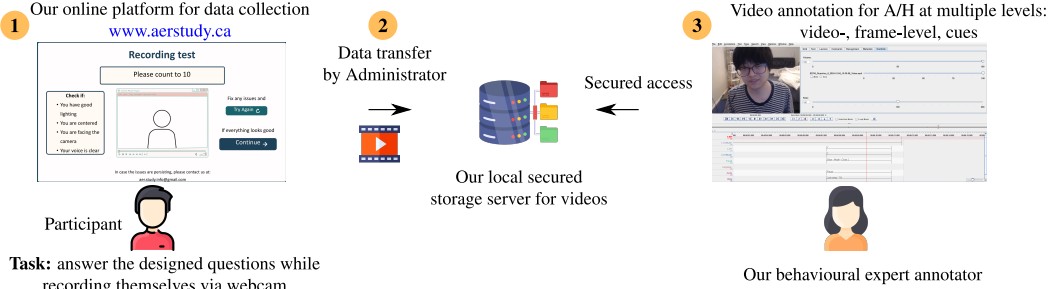

Figure 2: `BAH` dataset collection and annotation procedure. First, participants access our web platform. They undergo an initial test/calibration to ensure the quality of the data. An avatar guides them throughout the entire process. Seven questions are presented to each participant. They are recorded while answering them. Once the data is captured, it is transferred by the Administrator to our local server. It is then annotated at several levels by experts to determine when A/H occurs.

tion" (AER) web-based platform (www.aerstudy.ca) where participants could record their responses to specific questions using their own computers or devices with a camera and a microphone. Users received secure credentials to access the data collection platform, or they created their own account. Participants first complete a brief survey to provide demographic information and indicate consent preferences (e.g., inclusion in secure datasets, challenges, or publications). They were then redirected to the AER platform, where they tested their camera and microphone and chose an avatar to interact with during data capture. The avatar guides them through seven questions. The session took approximately 30 minutes. Participants were recruited and compensated through the Prolific company (www.prolific.com), which also ensures population diversity and allows submission processing.

Participants answered seven questions designed by our behavioural team (Table 1), each one intended to elicit neutral, positive, negative, ambivalent, willing, resistant, and hesitant answers. Once the question was presented, the recording of the participant's response started. Skipping questions was allowed. At the end of each question, the participant has the option to rate their emotional response using a 5-point Likert-like scale. This self-rating was only employed for our analysis and did not serve as an annotation. The order of the questions was randomized. In addition, participants were not aware of what each question was expected to elicit as emotion. During this capture procedure, several pieces of information were gathered, including contact information of the participant, their demographics, consent, video recordings, survey responses, and software usage data (such as the time spent on each question). The software used to interact with the participants and capture their data was managed by the MBMC and the Concordia research team. The participants' data was systematically downloaded and transferred to a local secure server storage (located at ETS Montreal) by the team for annotation and further analysis.

The study obtained human ethics approval from both Concordia University (www.concordia.ca) (certification number: 30019002) and ETS(www.etsmtl.ca) (certification number: H20231203), and the study followed all standard ethical practices. The dataset was collected by researchers from Concordia University and the Montreal Behavioural Medicine Center (MBMC) in collaboration with the Laboratoire d'imagerie, de vision et d'intelligence artificielle (LIVIA) at ETS Montreal. The dataset was collected between September 2024 and April 2025 in batches. This allowed us to adjust the targeted population (regarding participants' sex and Canadian province of residency) to ensure the dataset diversity.

We note that while the `BAH` dataset is sourced exclusively from Canadian participants, its sample collection was designed to be representative in terms of province and sex, two key demographic variables relevant to health behaviour research in Canada. Additionally, the dataset includes individuals from a wide range of backgrounds, including different ethnicities and national origins, which reflects the cultural diversity of the Canadian population. This diversity, often lacking in many existing datasets, enhances the relevance and applicability of our findings within the Canadian context. Despite the geographic limitation, we consider this dataset a meaningful starting point for

understanding ambivalence and hesitancy in health-related behaviour. We plan to expand to other countries and cultural settings to support broader generalizability.

**Annotation**. Three annotators were trained in expression recognition, specialized in identifying A/H, and in the annotation process of audio-visual data. A two-stage process was used: first, a global-level annotation determined the presence of A/H in each video; then, a frame-level annotation identified the precise segments where A/H occurred, specifying the start and end times (i.e., onset and offset) of each instance. Annotators also provided certainty ratings, and for some segments, indicated the cues that supported their judgment. To identify A/H, annotators tracked expressions across well-established modalities (facial expressions, body language, audio, and language) and flagged cases where inconsistencies between modalities were observed. Each A/H segment has a timestamp start/end in addition to its own cues and modalities determined by an annotator, allowing for better precision. Multiple A/H segments could be present within a video. We do not include an "apex" nor a continuous annotation, as ambivalence and hesitancy do not reliably exhibit a peak moment of maximum intensity. Instead, they tend to manifest as sustained or fluctuating states, making the concept of an apex incompatible with their typical temporal structure. The videos were annotated following a codebook created specifically for the study. Videos were annotated using the ELAN 8 (archive.mpi.nl/tla/elan) software (Figure 2).

The annotation process followed a structured training protocol supported by a detailed training manual. Annotators first received a conceptual introduction to A/H, followed by hands-on training in using the ELAN annotation software. Practical application was conducted using a standardized set of videos from the dataset. This phase also introduced annotators to the codebook, emphasizing the cue list, with examples spanning facial, vocal, verbal, and bodily expressions. Annotators received feedback, and additional sessions were provided when further alignment was needed. Only after this training phase, and a final assessment, did annotators proceed to independent annotation. To ensure large labeled data, `BAH` was divided among the three annotators, where each video is labeled by one annotator. However, we analyzed over 10 random videos with a multi-annotator setup to assess inter-annotator agreement. We considered Fleiss's Kappa measure (Fleiss, 1971) for more than 2-annotator cases. At the global level, the score measure was 0.65 (substantial agreement), while at the frame level, we had 0.41 (moderate agreement). When considering only videos where all three annotators agreed they had A/H, the score was 0.50.

To promote consistency, annotators were instructed to flag cases of uncertainty or complexity. These cases were discussed collaboratively, often through co-annotation. A consistent lead annotator facilitated resolution efforts, ensuring that decisions reflected a shared interpretation. In parallel, a comprehensive annotation protocol guided how videos were managed, accessed, and annotated. Annotators followed standardized procedures: (1) watch the video without taking notes to understand the participant and context; (2) re-watch the video to identify A/H segments and record start and end times; (3) reassess and refine selected segments; (4) identify and assign cues using the codebook; and (5) if needed, watch the video without audio or visual elements to isolate specific signals. Anno-

| Question no. | Response | Prompt |
|---|---|---|
| 1 | Neutral | Tell us about an activity you commonly do after waking up. |
| 2 | Positive | Talk about an activity that brings you joy, for example, a hobby. Tell us why. |
| 3 | Negative | Talk about an activity you dislike doing, for example, a chore or something you find boring or annoying. Tell us why. |
| 4 | Ambivalent | Tell us about something you enjoy doing but wish you stopped doing (like a guilty pleasure) or something you don't do but wish you did. |
| 5 | Willing | Tell us about an activity you are almost always willing to do, for example, with friends, at work, at home. |
| 6 | Resistant | Tell us about something people around you do, but that you would not be willing to do, for example, with friends, at work, at home. |
| 7 | Hesitant | Tell us about something you could have done already but haven't done yet, for example, something you are procrastinating or haven't made up your mind about. |

Table 1: The 7 questions (prompts) designed by our behavioural science experts to capture videos for the `BAH` dataset. To avoid influencing the participant's answers, they are only shown prompts without indicating the expected emotion/response.

tators were also encouraged to consult other videos from the same participant to establish expressive baselines in ambiguous cases.

The presence of A/H is assigned a single label (1), while its absence is assigned the label 0. Each video has a global- and frame-level label which can be used to train and evaluate ML models. The provided cues can also be used for the interpretability aspect, as well as to build insights into how people express A/H. The dataset is structured participant-wise, which can also be useful for personalization training scenarios. The `BAH` dataset that is being made public contains only videos of participants who consented for their data to be made public.

## 2.2 DATASET VARIABILITY

The `BAH` dataset is designed to approximate the demographic distribution of sex and provincial representation in Canada. It is composed of 300 participants across Canada from nine provinces, where 25.7% of participants are from British Columbia, followed by Alberta with 19.7% and Ontario with 17.3% [1] . All participants agreed to be part of this dataset. However, 61 participants (20.3%)[2] did not consent to be in publications, while only seven participants (2.3%) did not consent to be part of challenges. The majority of recorded videos are in the English language, and very few are in the French language. Each participant can record up to seven videos, and 113 participants have recorded the full seven videos. We obtained an average of $\sim 4.75$ videos/participant where each participant has an average of $\sim 2.59$ videos with A/H which is equivalent to $\sim 520.85$ frames of A/H (or $\sim 21.49$ seconds of A/H). The dataset amounts a total of 1,427 videos ($\sim 10.60$ hours) where 778 videos contain A/H ($\sim 1.79$ hours). This amounts to 916,618 total frames, where 156,255 contain A/H. Since captured videos represent answers to questions, they are relatively short. `BAH` dataset has an average video duration of $26.76 \pm 16.47$ (seconds) with a minimum and maximum duration of 3 and 96 seconds.

An important characteristic of `BAH` is the duration of A/H segments in videos. `BAH` dataset counts a total of 443 videos with multiple A/H segments and 332 videos with only one A/H segment. In total, there are 1,504 A/H segments. In particular, the duration of segments varies, but it is brief with an average of $4.29 \pm 2.45$ seconds, which is equivalent to $103.89 \pm 58.70$ frames. The minimum and maximum A/H segment is 0.004 seconds (1 frame) and 23.8 seconds (572 frames), respectively.

In terms of participants' age, the dataset covers a large range from 18 to 74 years old. In particular, 37.7% of the participants cover the range 25-34 years, followed by the range of 35-44 years with 24.3%, then the range of 18-24 years with 20.7%. In terms of sex, 52.0% are female, while 47.3 are male. As for ethnicity variation, 54.0% of the participants identified as "White", followed by "Asian" with 21.0%, and "Mixed" with 10.7%, then "Black" with 9.7%. The majority of the participants were not students (67.0%), which limits the common issues in recruit bias in other datasets.

The public `BAH` dataset contains the row videos, detailed A/H annotation at video- and frame-level, timestamps of A/H, cues, script to extract all frames, and per participant demographic information, including age, birth country, Canadian province where the participant lived, ethnicity, ethnicity simplified, sex, student status, consent to use recordings in publications. More details about the dataset diversity are provided in the appendix.

## 2.3 ETHICAL CONSIDERATION, DATASET ACCESSIBILITY AND INTENDED USES

The collected data of human participants closely follows standard ethical considerations. The project to collect `BAH` data was approved by ethical committees from both collaborating universities, Concordia University under agreement n° 30019002 and ETS under agreement n° H20231203. Once recruited, participants had access to the full consent form prior to accessing the data capture platform and starting their data capture procedure. They were provided with details of the study, as well as a list of the potential risks and benefits of participating in the study. They were asked to read the consent form thoroughly, and they were provided with a clear and simple video that summarized the consent form. Participants were then able to decide the type of access they wanted researchers to have to their audiovisual data, including whether they wanted their images to be used for publica-

---

[1] Provinces: 'Manitoba (MB)', 'Alberta (AB)', 'Nova Scotia (NS)', 'Newfoundland and Labrador (NL)', 'Saskatchewan (SK)', 'New Brunswick (NB)', 'Ontario (ON)', 'Quebec (QC)', 'British Columbia (BC)'.

[2] The list of these participants is provided within the shared files of the `BAH` dataset.

tions and presentations. In addition, these options were presented again at the end of the data capture procedure, just in case they changed their minds about their participation in the study or the use of their data after they had finished recording their responses. At the end of the study, participants receive, via email, a copy of the consent form that included their choices about data usage and the contact information for the team should they have any further questions. Participants were given numerical codes for anonymity.

Following the guidelines of the funding agency of the Canadian government, the BAH dataset is made public with open credentialed access for research purposes. To access the dataset, users are required to fill in a request form and sign an End-User License Agreement (EULA) as commonly done to ensure dataset security. Upon access approval, the user will receive a link to download the full dataset, including row videos, detailed annotation, cues, participants' metadata, cropped-and-aligned-faces, and audio transcripts. BAH uses a proprietary license for research purposes. The dataset is hosted on a secure server at ETS as it is intended for long-term availability. Our public code github.com/LIVIAETS/bah-dataset is under an open-source license (BSD-3-Clause license). The code website will be used as a permanent page for the dataset that will reflect any future updates. Despite all our precautions, our dataset may still be misused. We consider a thorough review of requests before granting data access.

Part of the BAH dataset has already been used by the public for the Ambivalence/Hesitancy (AH) Recognition Challenge[3], organized in the "8th and 10th Workshop and Competition on Affective & Behavior Analysis in-the-wild (ABAW) in conjunction with CVPR 2025 and 2026" (Hallmen et al., 2025; Kollias et al., 2025; Savchenko, 2025). A similar access protocol was implemented.

Our primary goal of building the BAH dataset was to make public a first and unique dataset for A/H recognition in videos. Given the content of the dataset, its multimodal aspect, and the provided annotations, it can be used to train and evaluate ML models for A/H recognition in videos at frame- and/or video-level with different learning scenarios. Since data is individually collected from participants, it can also be used for model personalization using domain adaptation. The cues used by annotators can also be used to learn interpretable models, providing further insight into our understanding of A/H in human behaviours. Such understanding and recognition of A/H can be leveraged in downstream tasks such as behavioural change, interventions and recommendations in clinics or through automated systems such as virtual trainers/assistants.

## 2.4 EXPERIMENTAL PROTOCOL

**Dataset split**. The dataset is divided randomly based on participants into 3 sets: train (195 participants), validation (30 participants) and test (75 participants) sets. We ensured that the 3 splits represent the total data distribution. The train and validation sets amount to 3/4 of the total participants, while 1/4 goes to the test set. Videos of one participant belong to one and one set only. The details of each set are presented in Table 2. The split files are provided along with the dataset files. They contain the split in terms of videos and frames, ready to use. Note that the dataset is highly imbalanced as depicted in Table 3, especially at the frame level, where only 17.04% contains A/H. This factor should be accounted for during training and evaluation. The dataset can be used for training at the video- and/or frame-level. The participant identifiers are provided in the splits, allowing subject-based learning scenarios.

| Data subsets | Train | Validation | Test | Total |
|---|---|---|---|---|
| Number participants | 195 | 30 | 75 | 300 |
| Number participants with A/H | 144 | 27 | 75 | 246 |
| Number videos | 778 | 124 | 525 | 1427 |
| Number videos with A/H | 385 | 75 | 318 | 778 |
| Number frames | 501,970 | 79,538 | 335,110 | 916,618 |
| Number frames with A/H | 76,515 | 13,984 | 65,756 | 156,255 |
| Total duration (hour) | 5.80 | 0.92 | 3.87 | 10.60 |
| Total duration with A/H (hour) | 0.87 | 0.16 | 0.75 | 1.79 |

Table 2: BAH dataset split into train, validation, and test sets.

---

[3]https://affective-behavior-analysis-in-the-wild.github.io/8th/#counts3

| Data subsets | Train (%) | Validation (%) | Test (%) | Total (%) |
|---|---|---|---|---|
| Participants with A/H | 73.85 | 90.00 | 100.00 | 82.00 |
| Videos with A/H | 49.49 | 60.48 | 60.57 | 54.51 |
| Frames with A/H | 15.24 | 17.58 | 19.62 | 17.04 |
| Duration with A/H | 15.09 | 17.41 | 19.44 | 16.88 |

Table 3: Imbalance rate of BAH dataset split across train, validation, and test sets: (Total # items with A/H)/(Total # items).

**Evaluation metrics**. We refer here to the positive class as the class with label 1, indicating the presence of A/H, while the negative class is the class 0, indicating the absence of A/H. To account for the imbalance in the BAH dataset, we use adapted standard evaluation metrics: - Average F1 (AVGF1) score, which is the unweighted mean of F1 of the positive and negative classes. - Average precision score (AP) of the positive class, which accounts for the performance sensitivity to the model's confidence. For AP score, a threshold list between 0 and 1 is used with a step of 0.001. Evaluation code of all measures is provided along with the public code of this dataset.

## 3 BASELINE RESULTS

This section provides preliminary results of different baseline models on our BAH dataset. In particular, we provide the performance of models for the supervised frame-level classification task. We consider a 2-class classification problem where each frame is annotated, and models predict two outputs: one for the positive class (presence of A/H), and a second for the absence of A/H. Supervised video-level classification performance is included in the appendix in addition to other tasks – zero-shot prediction, and personalization through unsupervised domain adaptation.

We initially focus on the impact of using single vs multimodal learning for frame-level classification. Then, the performance of different individual modalities are explored, along with their multimodal fusion. In addition, we investigate the impact of temporal modelling and context vs single frame learning. Next, we present the pre-processing of the three different modalities used: visual (facial), audio (vocal), and text transcripts (textual), and describe the baseline models used in each case.

### 3.1 PRE-PROCESSING OF MODALITIES

**1) Visual**. All frames from each video are extracted, and for each frame, faces are located using the RetinaFace model (Deng et al., 2019), cropped, and then aligned. The face with the highest score is stored in case multiple faces are detected in a frame. Faces are resized to $256 \times 256$ and stored as RGB images with a file name that maintains the order of frames. The video frame rate is 24 FPS. **2) Audio**. We follow standard procedure to process audio data (Praveen & Alam, 2024b; Richet et al., 2024; Zhang et al., 2023b). For audio modality, we first convert videos to single audio channels (mono) with a 16k sampling rate into wav format. The log melspectrograms features are extracted using the VGGish model (Hershey et al., 2017)(github.com/harritaylor/torchvggish). A hop of 1/FPS of the raw video is used to extract the spectrograms to synchronize audio with other modalities. **3) Text**. The collected data captures the audio of participants. We consider audio transcripts as an extra modality that can help in recognizing A/H since text is a significant cue used by annotators. To this end, we transcribe the audio of each video and detect the language using the Whisper model (Radford et al., 2023) (Whisper large-v3 multilingual: huggingface.co/openai/whisper-large-v3). We provide the timestamp of each transcript. Word-level features are then extracted using the BERT Base Uncased model (Devlin et al., 2019)(pypi.org/project/pytorch-pretrained-bert/). A word may span more than one frame. To synchronize with other modalities, a word-level feature is repeated per its timestamp for all the frames that correspond to the word.

Note that both cropped and aligned faces, and video transcripts are shipped with the shared public BAH dataset. Researchers can choose to use them or build their own.

### 3.2 PRE-TRAINING OF VISUAL BACKBONE

For audio and text modality, features are extracted offline and stored as described above. For visual modality, we explore different architectures, including CNN- and ViT-based(Dosovitskiy et al.,

|  | Without context | | With context (TCN) | |
|---|---|---|---|---|
| Backbone | AVGF1 | AP | AVGF1 | AP |
| APViT (Xue et al., 2022) | 0.5051 | 0.1906 | 0.5019 | 0.2069 |
| ResNet18 (He et al., 2016) | 0.5074 | 0.1940 | 0.5079 | 0.1993 |
| ResNet34 (He et al., 2016) | **0.5138** | 0.1952 | 0.4998 | 0.1984 |
| ResNet50 (He et al., 2016) | 0.4737 | 0.1942 | 0.4985 | 0.1915 |
| ResNet101 (He et al., 2016) | 0.4929 | **0.1967** | **0.5165** | **0.2070** |
| ResNet152 (He et al., 2016) | 0.4889 | 0.1843 | 0.5084 | 0.2058 |

Table 4: Performance of the visual modality (facial expressions) on the test subset of `BAH` for frame-level classification. It indicates the impact of architecture and context.

2021). In particular, we explore the ResNet family, including ResNet18, 34, 50, 101, and 152 (He et al., 2016). For the ViT family, we consider a recent model designed for basic emotion recognition, APViT (Xue et al., 2022). First, we pre-train each model on a basic emotion recognition task, including these emotions: "Anger", "Disgust", "Fear", "Happiness", "Sadness", "Surprise". To this end, we collected a large mixed dataset composed of 3 public common datasets for emotion recognition using images: `RAF-DB` (Li et al., 2017), `AffectNet` (Mollahosseini et al., 2019), and `Aff-wild2` (Kollias & Zafeiriou, 2019). This amounts to more than 0.54 million training images. Models are trained for basic emotion classification for 60 epochs with a batch size of 1,424 samples using 4 parallel NVIDIA A100 GPUs with 40 GB of memory. Standard cross-entropy loss and Stochastic Gradient Descent (SGD) are used for training. Once pretrained, each model is further fine-tuned on our `BAH` train set for A/H recognition. To account for class imbalance, we perform under-sampling of the negative class over the training set. This is achieved by randomly sampling the negative class samples to be the same as the positive class. The weights of both models are made public. The backbones of each model are used later for feature extraction of the visual modality. We compared variant visual modalities, including cropped faces, full-frame, and head-pose. Results show that cropped faces yield better results. Details are included in the supplementary materials.

## 3.3 IMPORTANCE OF CONTEXTUAL LEARNING

In this section, we aim to answer the question of whether context modelling can help better A/H recognition. This is particularly interesting since A/H does not occur instantly, but within a context. Text and audio modality already capture context in their features, making using them to answer this question less efficient. However, we can obtain frame features with and without any context. Therefore, we consider visual modality with different backbones to answer our question.

In the case of modelling without context, models simply train on independent frames without considering any context or dependency between them. Inference is done in the same way. In the case of modelling with context, both training and inference leverages temporal dependency between frames. To this end, a window of adjacent frames is fed to the model. We then use a temporal convolutional network (TCN) (Bai et al., 2018) after the visual backbone to capture relations between frame embeddings. The window length defines the extent of the context. Table 4 shows the obtained results. Regardless of the context, we observe a very low performance over `AP`, highlighting the difficulty of recognizing A/H based on images alone. In particular, `AP` is below 0.2070. On the other hand, `AVGF1` reaches 0.5138. As a reference, predicting every frame as the

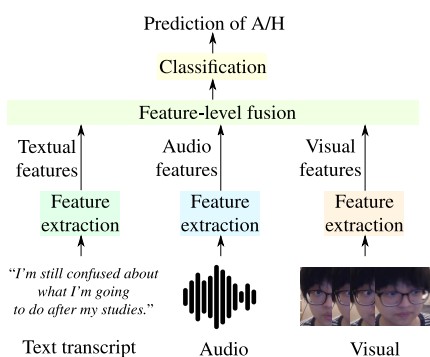

Figure 3: Multimodal model used to produce baseline performance (Richet et al., 2024).

negative class yields an `AVGF1` of 0.4459. Overall, using context boost performance of all metrics across all architectures. This is expected as A/H does not usually occur at a single frame but within a context. This makes its recognition from a single frame challenging. We recall that the average A/H segments span 103 frames (or 4.29 seconds). Future work should account for the temporal context for better performance. However, as we will show in the ablations in the appendix, a very large context could lead to poor performance. Note that large ResNet models seem to yield better performance overall. Unless mentioned otherwise, all our next experiments will use ResNet101.

### 3.4 MULTIMODAL BASELINES

Since A/H is multimodal data, we explore the impact of using different modalities, including visual, audio, and transcript, using the model shown in Figure 3 (Richet et al., 2024). Results are reported in Table 5. Using the text modality alone yields better performance, `AP` of 0.2510 and `AVGF1` of 0.5497, compared to visual or text modalities over both metrics. Combining a pair of modalities improves the performance to 0.5586 for `AVGF1`, and 0.2609 for `AP`, in the case of audio and text. Combining the three modalities slightly reduces performance in terms of `AP`, suggesting that more adapted fusion techniques are needed to recognize affect conflicts between/within modalities.

| Modalities | `AVGF1` | `AP` |
|---|---|---|
| Visual | 0.5165 | 0.2070 |
| Audio | 0.4658 | 0.2238 |
| Text | 0.5497 | 0.2519 |
| Visual + Audio | 0.5205 | 0.2225 |
| Visual + Text | 0.5547 | 0.2479 |
| Audio + Text | **0.5586** | **0.2609** |
| Visual + Audio + Text | 0.5502 | 0.2548 |

Table 5: Frame-level classification performance of multimodal models on `BAH`. ResNet101 is used for the visual modality.

Table 6 shows the impact of using different feature fusing techniques including simple concatenation (CAN) (Zhang et al., 2023b), co-attention (LFAN) (Zhang et al., 2023b), transformer-based fusion (MT) (Waligora et al., 2024), and cross-attention fusion (JMT) (Waligora et al., 2024). We observe that the way of leveraging the interaction between the three modalities is a key factor. LFAN and CAN fusion lead over both metrics. Future works should pursue more adapted methods to A/H. Since A/H are usually expressed as a conflict between willingness and resistance, they can be perceived through a parallel affect conflict between modalities and or within modalities. For instance, a participant could say a sentence to convey a meaning but their facial expression, body behaviour, or tone may carry a contradictory emotion. Understanding such subtly and interconnection between different cues in different modalities could play an important role in designing robust methods for A/H recognition in videos.

We believe our new and unique dataset has brought a new, challenging research direction to better understand complex and subtle human emotions that is A/H. Given the multimodal nature of A/H, our `BAH` dataset provides an essential and valuable toolkit for the research community to design and evaluate their methods. Important key and critical downstream tasks could potentially benefit from these methods, including but not limited to clinical interviews, interventions, behavioural changes, and automated assistants such as online trainers. Our preliminary results suggest that leveraging context, multimodality, and their fusion could lead to better A/H recognition performance.

| Method | Fusion Approach | `AVGF1` | `AP` |
|---|---|---|---|
| LFAN (Zhang et al., 2023b) *(cvprw,2023)* | Co-attention | 0.5502 | 0.2548 |
| CAN (Zhang et al., 2023b) *(cvprw,2023)* | Concatenation | **0.5526** | **0.2631** |
| MT (Waligora et al., 2024) *(cvprw,2024)* | Transformer | 0.5137 | 0.2134 |
| JMT (Waligora et al., 2024) *(cvprw,2024)* | Cross-attention | 0.5241 | 0.2139 |

Table 6: Impact of feature fusion methods on test set of `BAH` at frame-level classification.

## 4 CONCLUSION

This paper introduces `BAH`, a new multimodal and participant-based dataset for A/H recognition in videos. It contains the videos of 300 recruited participants captured across 9 provinces in Canada. Participants recorded themselves using a webcam and a microphone through our web-platform while they answered 7 questions designed to elicit A/H. The dataset amounts to 1,427 videos for a total duration of 10.60 hours, with 1.79 hours of A/H. It was annotated by our behavioural science team at the video- and frame-level. Our initial benchmarking study yields limited performance, highlighting the difficulty of A/H recognition. Results also indicate that leveraging context, multimodality, domain adaptation and adaptive feature fusion are promising directions to improve the accuracy and robustness of ML models on `BAH`. Our dataset and code are made public.

The supplementary materials contain recommendations for future work about A/H recognition, related work, more detailed and relevant statistics about the dataset and its diversity, dataset limitations, implementation details, and additional results.

ACKNOWLEDGEMENTS

This work was supported in part by the Fonds de recherche du Québec – Santé (FRQS), the Natural Sciences and Engineering Research Council of Canada (NSERC), Canada Foundation for Innovation (CFI), and the Digital Research Alliance of Canada.

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

# Appendix

## Table of Contents

## A    RECOMMENDATIONS FOR FUTURE WORKS IN A/H RECOGNITION

The newly introduced A/H recognition task requires the model to be able to detect affect conflict cross-modalities in addition to within-modality. Standard multimodal models are trained to yield predictions aligned with the output supervision. This automatic training may focus on learning label patterns and miss acquiring a mechanism to understand affect conflict. To build more interpretable A/H recognition systems, we recommend a 2-level framework. The first level should focus on modelling affect per modality in an independent way. Off-the-shelf pretrained sentiment analysis (Sharma et al., 2025) models could be used. This first level should be separated from A/H since we can not detect it at modality level yet, at least for the cross-modality case. At the second level, a dedicated fusion mechanism should be used to assess whether there is affect conflict cross-modalities to make a decision. This module does a deeper work than simply fusing features as commonly done. It should acquire an understanding of affect conflict to be able to detect it. Such modular and interpretable framework allows introducing priors about affect conflicts.

Statistics extracted from annotators cues could be leveraged to constrain the model find conflicts cross-modalities. A specialized temporal modelling should be used to detect within-modality conflict based on the output of the first level. Statistics from A/H segments durations should be considered as A/H happens briefly. Context is also important to detect within-modality cases. Segmented body could help as well since it contains important cues and less noise compared to full frame. Our results showed that standard multimodal models and fusion technique yield modest performance. Future works should focus on designing specialized frameworks for A/H recognition.

## B    RELATED WORK

This section provides works in affective computing related to behavioural science.

## a) Affect Recognition using Machine Learning:

**Basic Emotions.** An important line for ML research in affective computing is discrete emotion recognition in facial image (facial Expression Recognition – FER) (Bonnard et al., 2022; Liu et al., 2024a; Kollias et al., 2025; Lee et al., 2023; Mao et al., 2024; Wang et al., 2024; Wu & Cui, 2023; Xue et al., 2021; Zeng et al., 2022; Zheng et al., 2023). This usually involves classifying facial images into one of seven or eight basic emotions, such as 'Happy', 'Sad', and 'Surprised'. Other works focus on videos (Liu et al., 2023a; 2021a;b; 2023b) as well. There has also a recent interest in designing robust FER methods that are interpretable (Belharbi et al., 2024a;b; Wang & Kawka, 2024; Xue et al., 2022). They typically produce a heat map that points to relevant regions used by a model to perform a prediction. This is usually formulated as an attention map or a Class-Activation Map (CAM) (Choe et al., 2022; Murtaza et al., 2025). Other work aims to predict ordinal levels (i.e, ordered labels), including pain and stress estimation (Aslam et al., 2024; Chaptoukaev et al., 2023; Zeeshan et al., 2024; Nasimzada et al., 2024); a task that can be extremely useful in healthcare applications. Some datasets such as BioVid (Walter et al., 2013) rely on advanced and expensive modalities such as bio-signals to predict pain for instance. Dimension recognition of emotions, typically aims to estimate continuous valence and arousal values linked to emotions (Dong et al., 2024; Praveen & Alam, 2024a; Praveen et al., 2023; 2021). Finally, another task related in emotion recognition is Action Units (AUs) detection (Jacob & Stenger, 2021; Luo et al., 2022). It aims to predicting active AUs in the face under a multi-label classification framework. Other works go further to estimate the intensity of AUs (Fan et al., 2020; Zhang et al., 2018), or both (Sánchez-Lozano et al., 2018), a much more challenging task.

**Compound Emotions.** Real-world scenarios often present complex emotions that combine basic ones. There has been recent interest in building affective computing models to predict compound emotions, a case where a mixture of basic emotions are expressed (Kollias, 2023; Richet et al., 2024). These are show in several practical real-world application since such complex emotions occur in daily interactions. However, they are more difficult to recognize as they are subtle, ambiguous, and resemble basic emotions. A recent specialized video-based dataset named C-EXPR-DB (Kollias, 2023) has been constructed for the design/evaluation of models. The dataset accounts for the difficulty of the task as different modalities are required to better recognize compound emotions.

Despite the recent progress in affect modelling, Ambivalence/Hesitancy recognition is still unexplored in ML. A possible reason is the lack of specialized dataset for training and evaluation of ML models. As it is implicated in healthcare and interventions, A/H is a common topic in behavioural science (Hohman et al., 2016; Manuel & Moyers, 2016). A/H recognition is related to compound emotion recognition task where intention and attitudes are conflicted or in a in-between state, between willingness and resistance (MacDonald, 2015), or positive and negative affect (Armitage & Conner, 2000). This can manifest in how an individual expresses them self and can be

| Dataset | Affect | Modalities | Subject-based | Num. of participants | Num. of samples | Environment | Annotation |
|---|---|---|---|---|---|---|---|
| RAF-DB (Li et al., 2017) | Basic/compound emotions | Images | No | – | 15,339 images | Wild | Image-level |
| AffectNet (Mollahosseini et al., 2019) | Basic emotions | Images | No | – | 450k images | Wild | Image label |
| Aff-wild2 (Kollias & Zafeiriou, 2019) | Basic emotions, Valence/Arousal, Action Units | Video, audio | No | – | 564 videos | Wild | Frame-level |
| MELD (Poria et al., 2019) | Basic emotions | Video, audio | No | – | 13000 utterances | Actors/TV-show | Frame-level |
| C-EXPR-DB (Kollias, 2023) | Compound emotions | Video, audio | No | – | 400 videos | Wild | Frame-level |
| UNBC-McMaster (Kollias & Zafeiriou, 2019) | Pain estimation | Frames | Yes | 25 | 200 videos | Lab | Frame-level |
| BioVid (Walter et al., 2013) | Pain estimation | Frames, biomedical signals (GSR, ECG, and EMG at trapezius muscle) | Yes | 90 | 18017 samples | Lab | Frame-level |
| RECOLA (Ringeval et al., 2013) | Apparent Emotional Reaction Recognition | video, audio, physiology (electrocardiogram, and electrodermal activity) | Yes | 46 | 46 videos | Lab | Frame-level |
| SEWA (Kossaifi et al., 2019) | Apparent Emotional Reaction Recognition | video, audio | Yes | 398 | 1,990 videos | Wild | Frame-level |
| WEMAC (Miranda Calero et al., 2024) | Discrete, dimensional emotions | Physiology (blood volume pulse, galvanic skin response, and skin temperature), audio | Yes | 100 | 100 records | Lab | Self-reported |
| StressID (Chaptoukaev et al., 2023) | Stress | EDA, ECG, Respiration, Face video, Speech | Yes | 65 | 587 videos | Lab | Frame-level |
| SchiNet (Bishay et al., 2019) | Estimation of Symptoms of Schizophrenia | video | Yes | 91 | 91 videos | Wild | Video-level |
| MESC (Chu et al., 2024) | Emotional Support Conversation | video, audio, text | Yes | – | 1,019 dialogues | Wild | Utterance-level |
| IEMOCAP (Busso et al., 2008) | Improvisations of scripted scenarios for basic emotions | video, audio, text | Yes | 10 actors | – | Lab/Actors | Frame-level |
| BAH (ours) | Ambivalence/Hesitancy | Video, audio, transcript | Yes | 300 | 1,427 videos | Wild | Video-level, Frame-level, A/H cues |

Table 7: Common affective computing datasets for emotion modelling in health contexts.

recognized (Hayashi et al., 2023) in their facial expression, tone, verbal, and body language. As a result, A/H exhibits a multimodal nature that comes as the result of subtle interconnection between different cues. Unfortunately, such discord is extremely difficult to spot; a task that requires human training. This is a tedious and expensive procedure, leading to ineffective and less scalable eHealth interventions under limited resources. Assisting healthcare providers with automatic, reliable and inconspicuous tools to help them recognize A/H can have a major impact in improving eHealth interventions.

Our BAH dataset fills in the gap in the literature, and to provide an important resource to design/evaluate ML models for A/H recognition task. It is a video Q&A dataset from which we extract audiovisual information with transcripts, offering multiple modalities. The dataset is fully annotated by behaviour science experts at video- and frame-level. In addition, cues used by annotators to recognize A/H at each segment are provided. This includes facial and vocal expressions, body language, language in addition to highlighting where there is inconsistency between the modalities. As shown in Table 7, our BAH dataset is competitive compared to existing affective computing datasets in terms of modalities, number and diversity of participants, and annotations. While no dataset matches the specific focus on A/H in digital health interventions, datasets like MESC (Chu et al., 2024), SchiNet (Bishay et al., 2019), and IEMOCAP (Busso et al., 2008) contain videos from interviews with psychological relevance. Therefore, BAH provides an important asset for the ML community to begin research in A/H recognition.

**b) Behavioural Science:**

**Health-Related Behaviour Change and Non-Communicable Diseases.** High-risk health behaviours, such as tobacco use, physical inactivity, unhealthy diets, and harmful alcohol consumption, are responsible for the vast majority of non-communicable diseases (NCDs), which include cardiovascular disease, type 2 diabetes, cancer, and chronic respiratory illnesses. According to the World Health Organization (WHO) (Ortiz et al., 2025), NCDs account for approximately 74% of global deaths, and these outcomes are disproportionately influenced by modifiable behavioural factors. Evidence suggests that around 80% of chronic disease risk is attributable to these high-risk behaviours.

Consequently, health-related behaviour change has become a primary target for preventive and therapeutic interventions. Traditional methods, such as motivational interviewing (MI) and cognitive behavioural therapy (CBT), rely on face-to-face clinical interviews, which remain foundational to behavioural health practice (O'Donnell et al., 2019). These interactions provide unique opportunities for clinicians to detect ambivalence, hesitancy, and other complex affective states, often through subtle verbal and nonverbal cues (Hall et al., 1995). Despite the growing shift toward digital platforms, clinical interviews remain the gold standard for eliciting meaningful emotional and cognitive responses, insights that are essential to tailoring behaviour change strategies. Efforts to change health behaviours over the long term are inherently complex. Individuals often experience ambivalence and hesitancy, understood as fluctuating between intention and resistance, when attempting to adopt healthier lifestyles. In traditional healthcare contexts, providers rely on both verbal communication and non-verbal cues (e.g., tone, gestures, facial expressions) to recognize and address such motivational conflicts. This in-person interaction allows for nuanced support that can adapt to a patient's readiness for change (Davidson & Scholz, 2020). The purpose of developing multimodal A/H recognition systems is to capture and replicate this nuanced understanding of patient behaviour within digital health interventions, thereby supporting clinicians and scaling behavioural health care.

**Multimodal Cues and the Detection of Complex Emotions.** Identifying complex emotional states such as ambivalence, resistance, or hesitancy is crucial for tailoring behavioural interventions. Research in psychology and human-computer interaction has shown that complex emotional states, such as ambivalence, uncertainty, or defensiveness, are communicated through a combination of facial expressions, body posture, vocal tone, speech patterns, and physiological responses (Guo et al., 2018; Pantic & Rothkrantz, 2003). In digital contexts, however, the absence of physical presence makes this task more difficult. Recent research in psychology and computer science has focused on the use of multimodal cues, such as facial expressions, voice tone, body posture, and physiological responses, as proxies for emotional and motivational states (Kraack, 2024; Yan et al., 2024). These cues can reveal underlying emotional conflict or uncertainty that might not be captured by self-report alone. Studies have shown that combining multiple input channels (e.g., audio-visual data) can enhance the accuracy of emotion recognition systems. For instance, multimodal datasets are being used to train models that detect affective states like confusion, frustration, and mixed emotions,

which are highly relevant in contexts such as education, mental health, and behaviour change. By incorporating these data streams, researchers can better approximate the nuanced human capacity for reading emotions, paving the way for emotionally aware systems (He et al., 2020; Zhao et al., 2021).

**Affective Computing and Personalized Digital Health Interventions** Affective computing, a sub-field of artificial intelligence (AI) focused on recognizing, interpreting, and responding to human emotions, holds promise for advancing personalized digital health interventions. By leveraging emotion-aware algorithms, digital platforms can better understand users' psychological readiness and tailor support accordingly (Lokhande et al., 2024; Vairamani, 2024). For example, interventions that dynamically respond to detected signs of resistance or disengagement may improve user retention and behavioural outcomes. Incorporating affective computing into digital health technologies also allows dynamic tailoring of content based on users' real-time affect, responsive dialogue, mimicking the adaptability found in face-to-face interactions. Recent advancements in conversational agents, voice analysis, and facial expression recognition have made it possible for digital interventions to adapt content delivery based on real-time emotional assessments (Khanna et al., 2022). This not only improves user engagement but also enhances intervention effectiveness by ensuring messages are delivered in an emotionally congruent and contextually appropriate manner (Hornstein et al., 2023).

## C  BAH DATASET LIMITATIONS

While BAH dataset offers a novel contribution to emotion recognition for digital behaviour change, several limitations should be considered.

**1) Data collection constraints.** The web-based platform occasionally experienced technical issues, preventing some participants from completing all seven videos. As participants used their own devices in home settings, video and audio quality varied significantly despite clear instructions and testing. Response length was participant-determined, leading to high variability in content. Some environmental noise or visual distractions (e.g., background conversations, movement) were present in a subset of recordings.

**2) Participant representation.** Although participants were recruited from nine Canadian provinces with diverse age and ethnic backgrounds, individuals from under-resourced areas or without reliable internet access were likely underrepresented. Gender identity was collected but not used in sampling, and no data on socioeconomic status was recorded. Digital literacy and access may have biased participation toward more tech-savvy individuals.

```
.
├── cropped-aligned-faces
├── Frames
├── split
├── split-frames
├── transcription
├── Videos
├── BAH_dataset_documentation.pdf
├── BAH_Dataset_EULA-2.pdf
├── bah-video.csv
├── extract_frames_from_videos.py
├── extract_frames_from_videos.sh
├── meta_data.yml
├── readme.md
├── version.txt
└── video_annotation_transcript.yaml
```

Figure 4: File structure of the shared BAH dataset.

**3) Multimodal and data balance issues.** The expressiveness of cues (facial, vocal, bodily, verbal) varied widely by participant, complicating consistent multimodal analysis. Though the dataset is balanced at the video level, frame-level imbalance exists (fewer A/H frames than non-A/H). Training strategies that account for class imbalance should be considered.

## D    BAH DATASET FILE STRUCTURE

Figure 4 shows the file structure of the shared BAH dataset. The file "BAH_dataset_documentation.pdf" contains the detailed documentation about all files/directory, including annotation structure.

## E    BAH DATA COLLECTION WEB-PLATFORM

Alongside the dataset files, we include a slide presentation of our "Automatic Expression Recognition" (AER) web-based platform (www.aerstudy.ca). The presentation is in the file "AER-web-platform.pdf". Figure 5 shows an example of the platform.

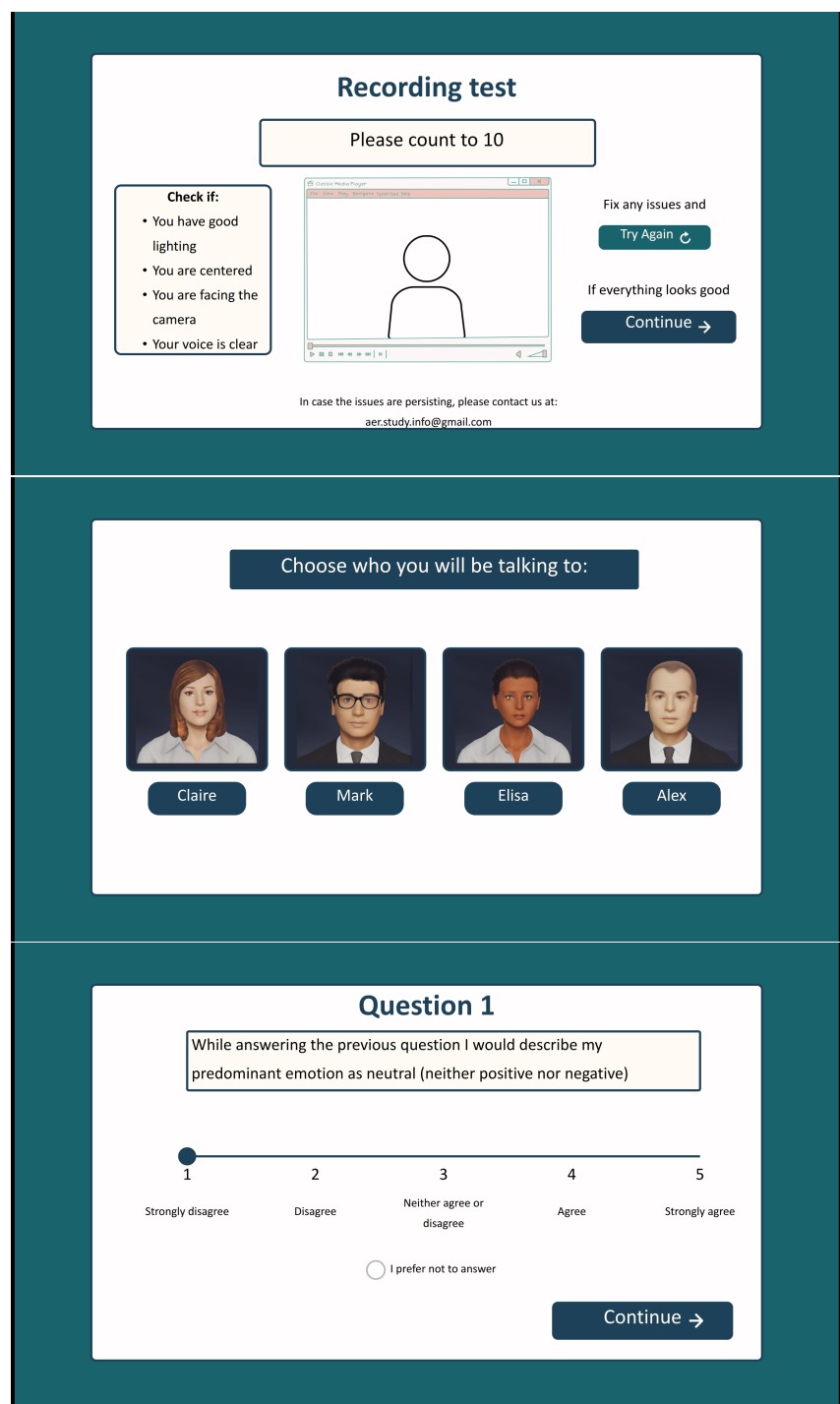

Figure 5: Examples taken from the platform to present our "Automatic Expression Recognition" (AER) web-based platform (www.aerstudy.ca).

## F    BAH DATASET DIVERSITY

This section includes more statistics about BAH dataset to highlight its diversity. Figure 11 shows a general overview via a nutrition label. Overall, BAH dataset has significant diversity. It covers different Canadian provinces, age range, ethnicities, and male/female presence. It has a large number

of videos (1,427) where 778 videos contain A/H. Most asked questions elicited A/H, especially question-4 (Ambivalent). In addition, since we have less control over the participants, and their environment, the dataset is considered in-the-wild. On top of video and audio modality, we provide audio transcript which has shown to be an important modality for A/H recognition. `BAH` is fully annotated at video- and frame-level. Moreover, annotators report the used cues to recognize A/H at each segment. All these properties make our dataset a realistic and relevant asset to design ML model for the task of A/H recognition in videos.

We include the following general information:

- Videos durations distribution (Figure 6a).
- Videos per participants distribution. (Figure 6b).
- Questions and A/H distribution (Figure 6c).
- A/H segments duration (Figure 7).
- Participants sex distribution (Figure 6d).

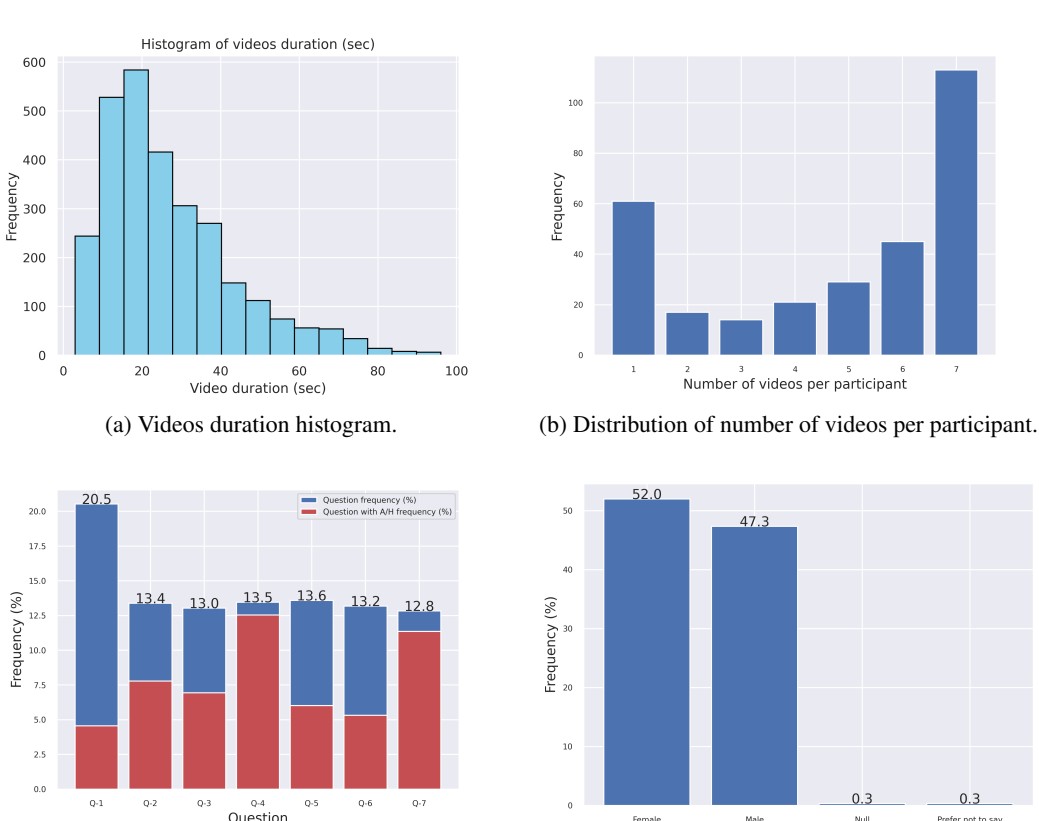

(a) Videos duration histogram.

(b) Distribution of number of videos per participant.

(c) Distribution over 7 questions: Num. videos per-question (blue), Num. videos with A/H (red).

(d) Sex distribution.

Figure 6: Video duration (a), and videos/participant (b), question distribution (c), and sex distribution (d) over `BAH` dataset.

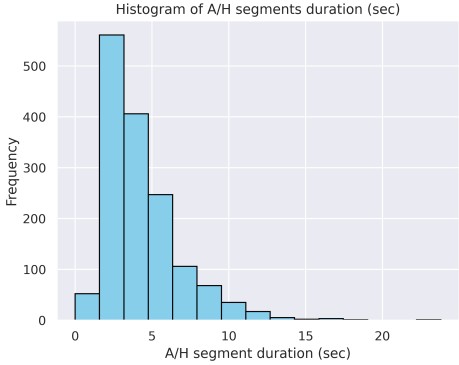

(a) Distribution of A/H segment duration in seconds.

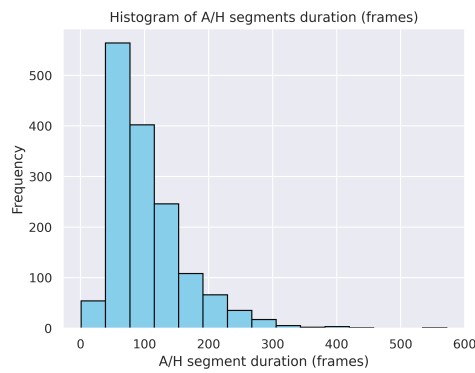

(b) Distribution of A/H segment duration in frames.

Figure 7: Distribution of A/H segment duration in seconds (a), and frames (b) over `BAH` dataset.

In addition, more demographics statistics are included as well:

- Participants' age distribution (Figure 8).
- Participants' age range distribution (Figure 9a).
- Distribution of Canada provinces where participants live (Figure 9b).
- Participants' simplified ethnicity distribution (Figure 10a).
- Participants' student-status distribution (Figure 10b).
- Participants' consent to use their data in challenges distribution (Figure 10c).
- Participants' consent to use their data in publications distribution (Figure 10d).
- Participants' birth country distribution (Table 8).

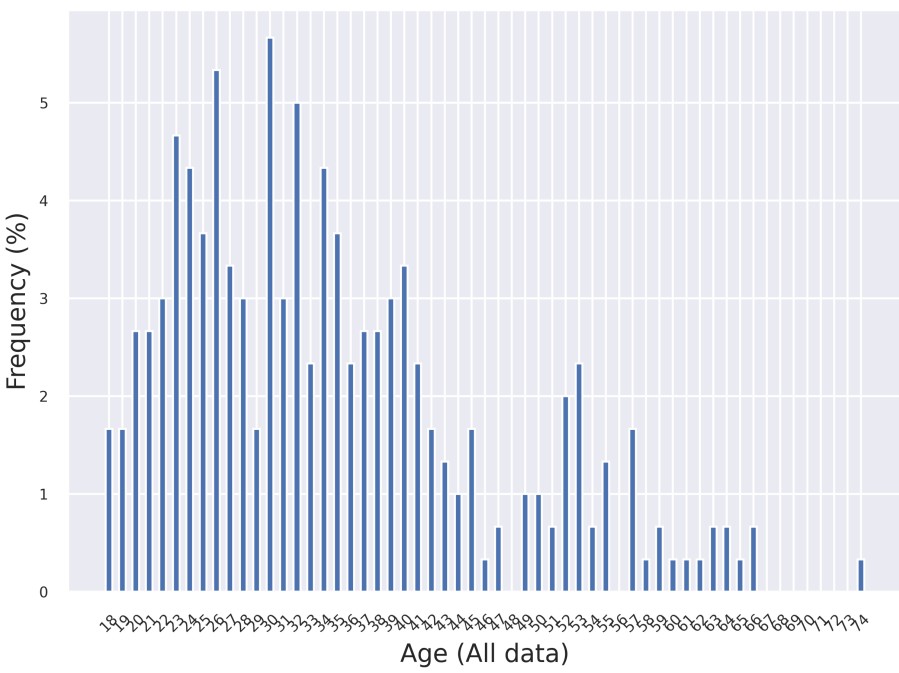

Figure 8: Participants' age distribution in `BAH` dataset.

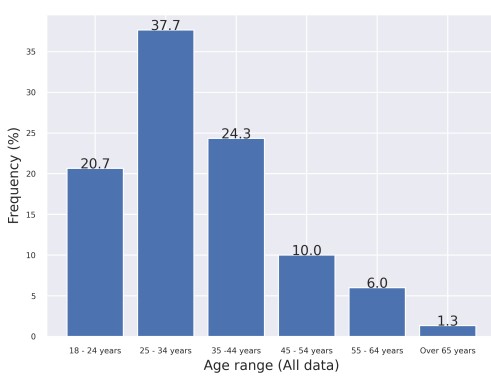

(a) Participants' age range distribution.

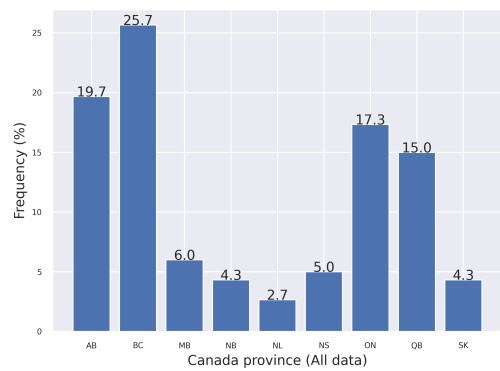

(b) Distribution of Canada provinces where participants live.

Figure 9: Participants' age range (a), and where the provinces where they live (b) over `BAH` dataset. Name of provinces: 'Manitoba (MB)', 'Alberta (AB)', 'Nova Scotia (NS)', 'Newfoundland and Labrador (NL)', 'Saskatchewan (SK)', 'New Brunswick (NB)', 'Ontario (ON)', 'Quebec (QC)', 'British Columbia (BC)'.

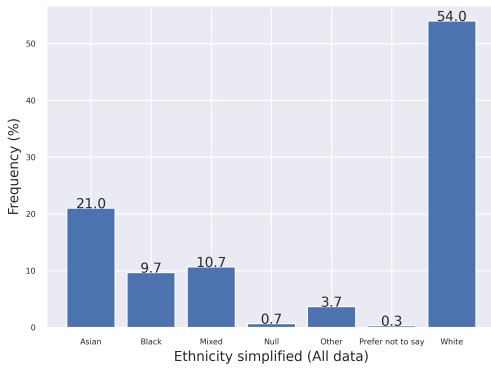

(a) Participants' simplified ethnicity distribution in BAH dataset.

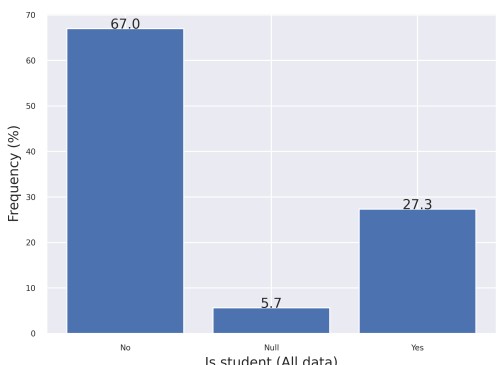

(b) Participants' student-status distribution in BAH dataset.

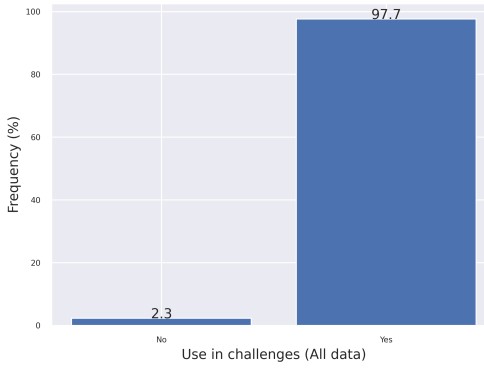

(c) Participants' consent to use their data in challenges distribution in BAH dataset.

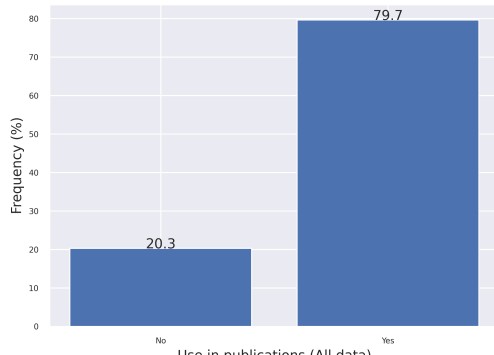

(d) Participants' consent to use their data in publications distribution in BAH dataset.

Figure 10: Distribution of participants' simplified ethnicity (a), their student-status (b), their consent to use their data in challenges (c), and publications (d) over BAH dataset.

| Birth country | Number of participants |
|---|---|
| Algeria | 1 |
| Australia | 1 |
| Bangladesh | 2 |
| Belize | 1 |
| Bulgaria | 1 |
| **Canada** | **127** |
| **China** | **8** |
| Colombia | 1 |
| France | 1 |
| Germany | 3 |
| Ghana | 1 |
| Hong Kong | 1 |
| India | 5 |
| Japan | 1 |
| Kenya | 1 |
| Macedonia | 1 |
| New Zealand | 1 |
| **Nigeria** | **9** |
| Null | 1 |
| Peru | 1 |
| Philippines | 6 |
| Russian Federation | 1 |
| Saint Lucia | 1 |
| South Africa | 1 |
| Sri Lanka | 2 |
| Taiwan | 1 |
| Thailand | 1 |
| Tunisia | 1 |
| Turkey | 2 |
| United Arab Emirates | 1 |
| United Kingdom | 6 |
| United States | 3 |
| Vietnam | 1 |

Table 8: Distribution of participants' birth country in BAH dataset. Top-3 countires are in bold.

# BAH Dataset Facts

**Dataset** BAH (Behavioural Ambivalence/Hesitancy – A/H)
**Nature of Dataset** A Dataset for Ambivalence/Hesitancy recognition in videos for participants recruited in Canada
**Participants Country** Canada
**Number of provinces in Canada** 9
**Provinces in Canada** 'Manitoba (MB)', 'Alberta (AB)', 'Nova Scotia (NS)', 'Newfoundland and Labrador (NL)', 'Saskatchewan (SK)', 'New Brunswick (NB)', 'Ontario (ON)', 'Quebec (QC)', 'British Columbia (BC)'.
**Number of participants** 300
**Number of videos** 1,427 where 778 videos contains A/H
**Average video length** $26.76 \pm 16.47$ (seconds) with a minimum and maximum duration of 3 and 96 seconds
**Total duration** 10.60 hours where A/H duration is 1.79 hours
**Total number of frames** 916,618 where 156,255 frames contains A/H
**Total number of A/H video segments** 1,504
**Length A/H video segment** $4.29 \pm 2.45$ seconds or $103.89 \pm 58.70$ frames. The minimum and maximum A/H segment is 0.004 seconds (1 frame), and 23.8 seconds (572 frames)
**Data capture web-platform** www.aerstudy.ca

Motivation

**Summary** Behavioural Ambivalence/Hesitancy (BAH) is a dataset collected for participant-based multimodal recognition of A/H in videos. It contains videos from 300 participants captured across 9 provinces in Canada, with different age, and ethnicity. Through our web platform, we recruited participants to answer 7 questions, some of which were designed to elicit A/H while recording themselves via webcam with microphone. BAH amounts to 1,427 videos for a total duration of 10.60 hours with 1.79 hours of A/H. Our behavioural team annotated timestamp segments to indicate where A/H occurs, and provide frame- and video-level annotations with the A/H cues. Video transcripts and their timestamps are also included, along with cropped and aligned faces in each frame, and a variety of participants meta-data.
**Original Authors** Manuela González-González, Soufiane Belharbi, Muhammad Osama Zeeshan, Masoumeh Sharafi, Muhammad Haseeb Aslam, Marco Pedersoli, Alessandro Lameiras Koerich, Simon L. Bacon, Eric Granger

Metadata

**URL** github.com/LIVIAETS/bah-dataset
**Keywords** Ambivalence/hesitancy, eHealth, digital health intervention, video, Deep Learning, Benchmark
**Available participants meta-data** Age, birth country, Canada province where the participant lives, ethnicity, ethnicity simplified, sex, student status, consent to use recordings in publications
**Video format** *.mp4
**Ethical Review** Concordia University under agreement n° 30019002 and ETS under agreement n° H20231203.
**License** Custom - for research purposes only.
**How to request the data?** Fill in this form, sign, and upload the EULA - https://www.crhscm.ca/redcap/surveys/?s=LDMDDJR3AT9P37JY
**First release** 2025

Annotation

**Annotators** 3 experts in behavioural science
**Video- and frame-level** Label "1" for presence of A/H, "0", its absence
**Cues provided by annotators for each A/H segment** Facial expressions, body language, audio and language in addition to highlighting where there is inconsistency between the modalities

Data size

**All files zipped** (*.zip) 9.6 GB

Figure 11: A data card styled (nutrition label) for BAH dataset.

# G   DEMOGRAPHICS ANALYSIS OF TRAIN, VALIDATION, AND TEST SETS

We provide in Figure 12 and 13 relevant distributions of the splits train (195 participants), validation (30 participants) and test (75 participants) sets in BAH dataset. The split participants-based where videos of participant belong exclusively to one split. We ensure that all splits have similar distribution in terms of 3 main factors that are deemed important in A/H variation across the population: gender, age, and ethnicity. We also include provinces distribution in Figure 14.

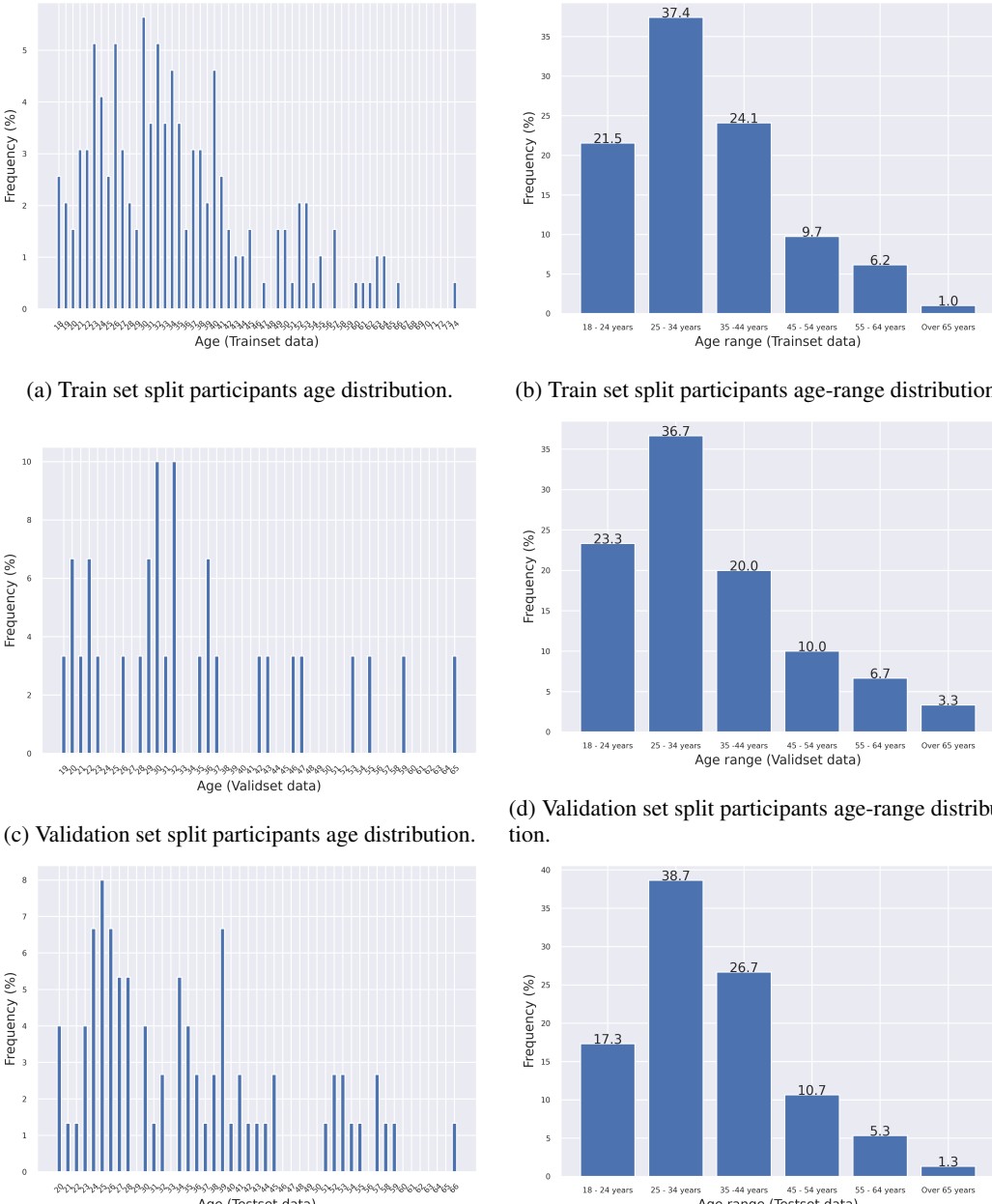

(a) Train set split participants age distribution.

(b) Train set split participants age-range distribution.

(c) Validation set split participants age distribution.

(d) Validation set split participants age-range distribution.

(e) Test set split participants age distribution.

(f) test set split participants age-range distribution.

Figure 12: Participants age and age-range distribution across all splits (train, validation, and test) in BAH dataset.

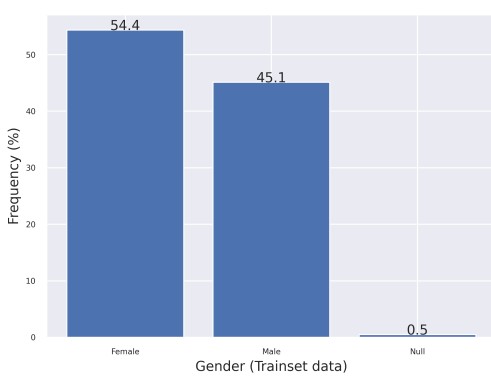

(a) Train set split participants sex distribution.

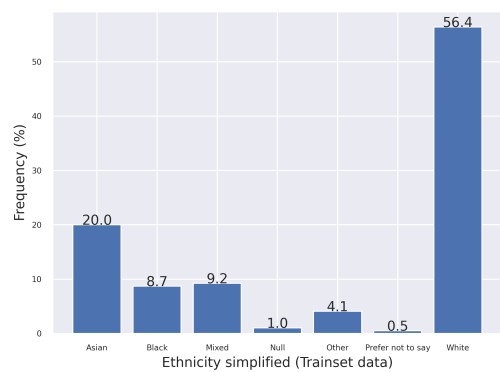

(b) Train set split participants simplified ethnicity distribution.

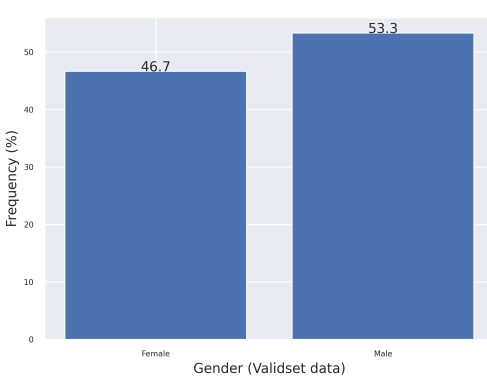

(c) Validation set split participants sex distribution.

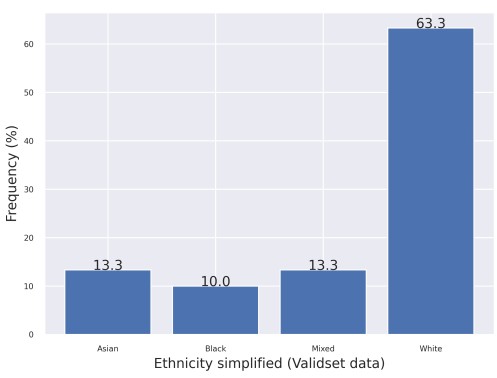

(d) Validation set split participants simplified ethnicity distribution.

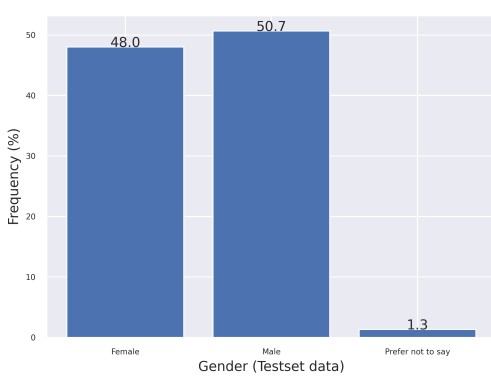

(e) Test set split participants sex distribution.

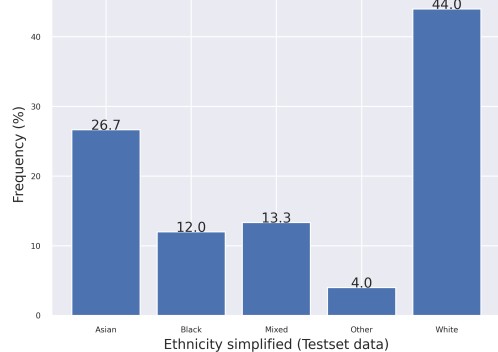

(f) test set split participants simplified ethnicity distribution.

Figure 13: Participants sex and simplified ethnicity distribution across all splits (train, validation, and test) in BAH dataset.

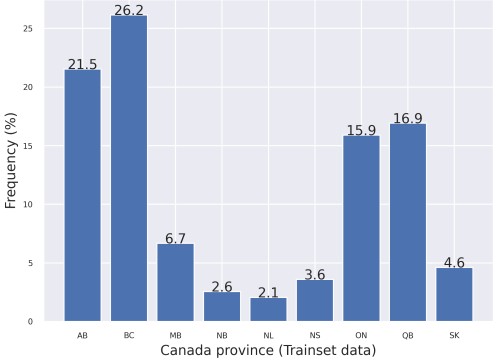

(a) Train set split participants provinces distribution.

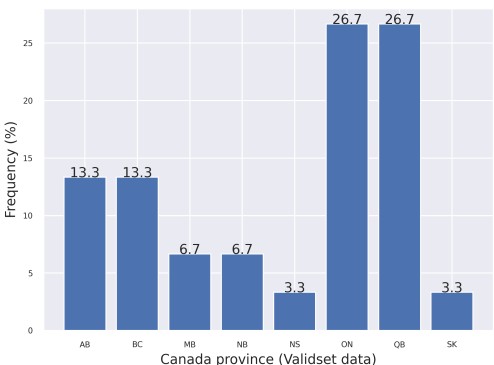

(b) Valid set split participants provinces distribution.

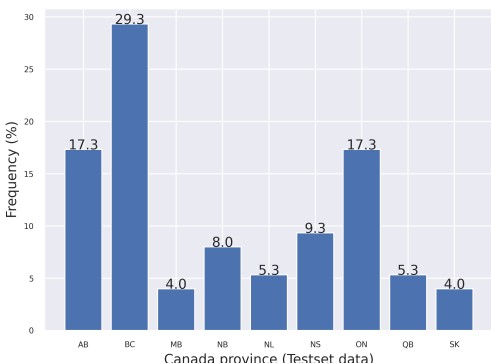

(c) Test set split participants provinces distribution.

Figure 14: Participants provinces distribution across all splits (train, validation, and test) in BAH dataset.

## H    ANALYSIS OF ANNOTATORS CUES

Figure 15a shows the distribution of modalities used by annotators to assess the presence of A/H. Overall, the four modalities contribution almost at the same rate in detecting A/H. Facial cues lead, followed by language, then body and audio. This suggests that the four modalities are equally important for A/H recognition. When looking to co-occurrence of these cues, we find that the four cues at once dominates by 54.1%, followed by "body-facial-language" with 14.7%, and "audio-facial-language". This suggest an adapted fusion of these four modalities is necessary to be able to assess there is an affect between them.

We include in Figure 16, 17 the distribution of cues used per modality. It shows that "Pause" is the dominate cues used in audio modality; "Filler sound" for language; "Gaze" for facial; and "Shake" for body.

We also analyzed the inconsistencies between modalities in Figure 18a, 18b. The inconsistencies between facial and language cues seem to be the lead by 42.6% of cases followed by language and body with 26.6%. This could be used as priors to recognize the case of A/H cross-modalities. When looking to co-occurrence, similarly, facial and language dominates alone with 21.4% of cases, followed by facial-language and language-body with the same rate. Leveraging these statistics by using them as constraints could help designing better A/H frameworks.

A/H can also be present within a single modality alone making it detection much more difficult compared to the case of cross-modality. Figure 19a shows that a large part of A/H cases fall into the case of within-modality, which is also spread across all questions (Figure 19b). A dedicated per-modality temporal modelling could help detect these cases.

These aforementioned statistics show that recognizing A/H requires considering all four modalities, and being able to detect affect conflict across pairs and higher combinations of modalities for the case of cross-modality. Additionally, a better temporal modelling is required to detect A/H in the case of within modality.

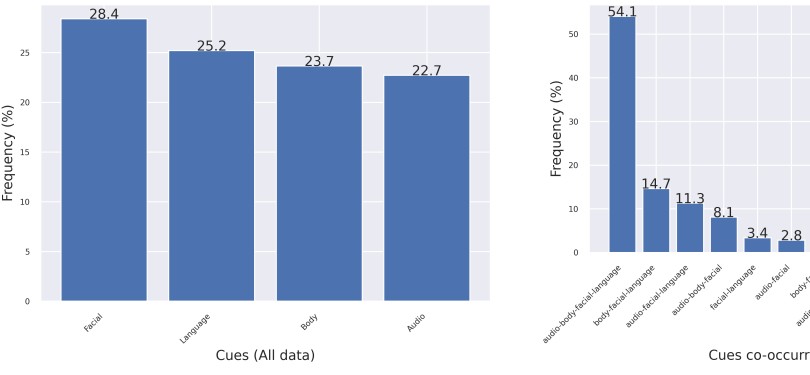

(a) Modalities cues distribution.  (b) Modalities cues co-occurrence distribution.

Figure 15: Modalities distribution used by annotators in BAH dataset.

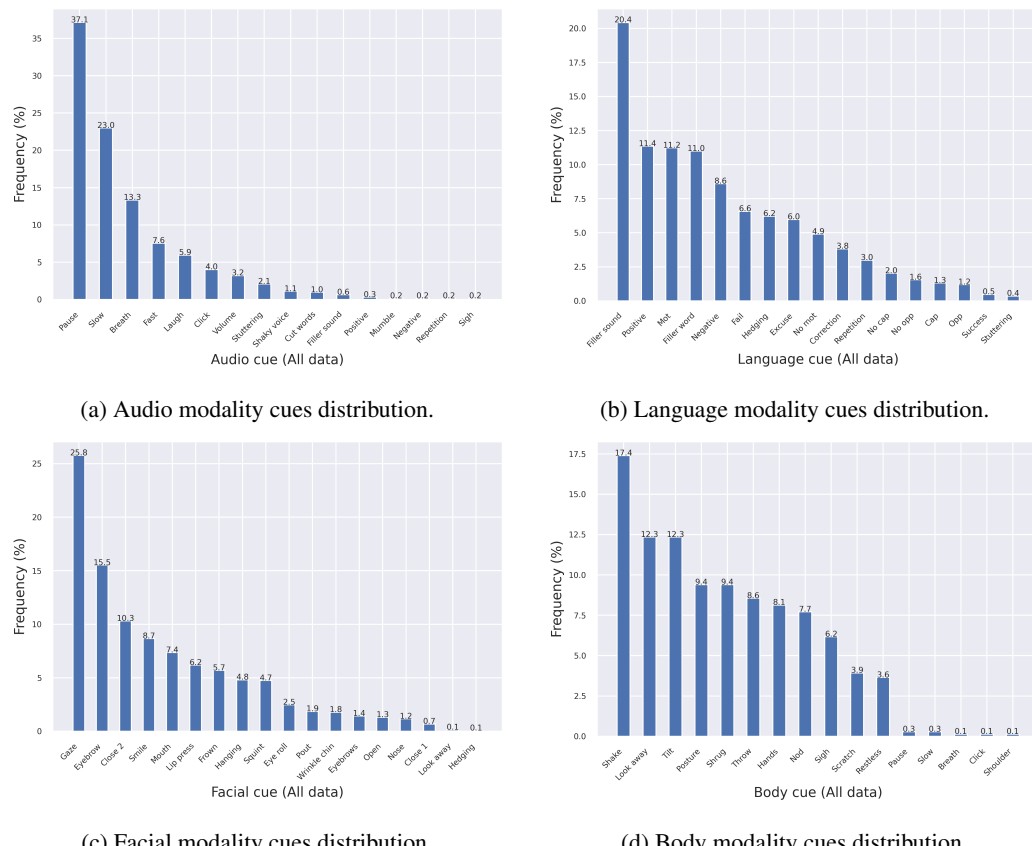

(a) Audio modality cues distribution.

(b) Language modality cues distribution.

(c) Facial modality cues distribution.

(d) Body modality cues distribution.

Figure 16: Per-modality cues distribution used by annotators in `BAH` dataset.

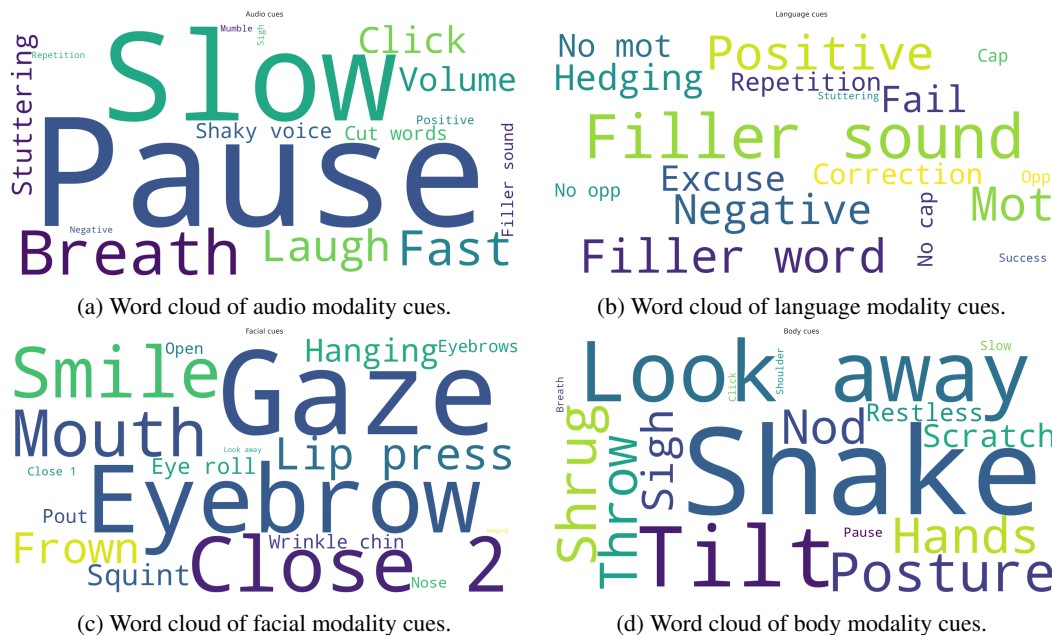

(a) Word cloud of audio modality cues.

(b) Word cloud of language modality cues.

(c) Word cloud of facial modality cues.

(d) Word cloud of body modality cues.

Figure 17: Word cloud of per-modality cues used by annotators in `BAH` dataset.

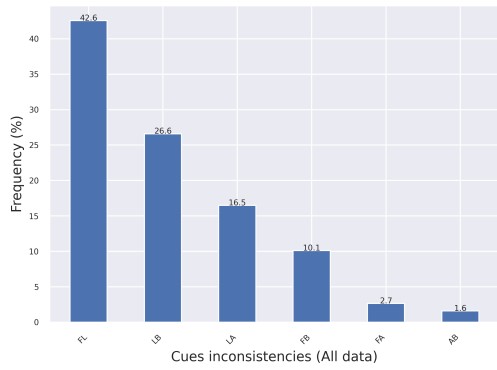

(a) Cues inconsistencies distribution.

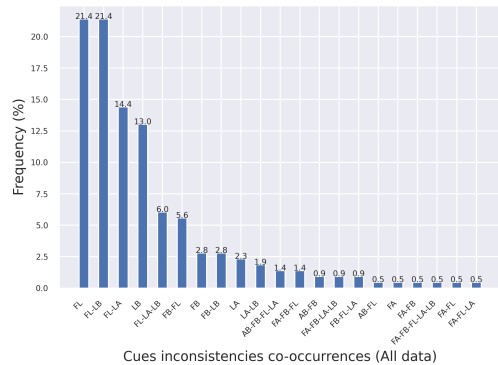

(b) Cues inconsistencies co-occurrence distribution.

Figure 18: Cues inconsistencies and their co-occurrence distribution used by annotators in BAH dataset.

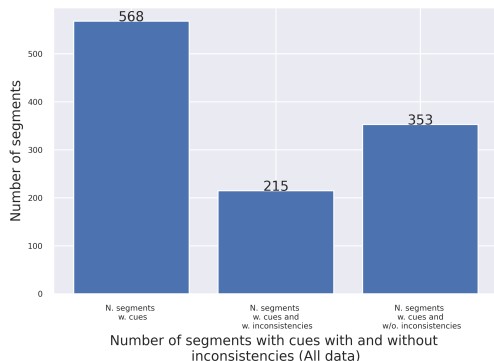

(a) Cues distribution with and without cross-modality inconsistencies over segments.

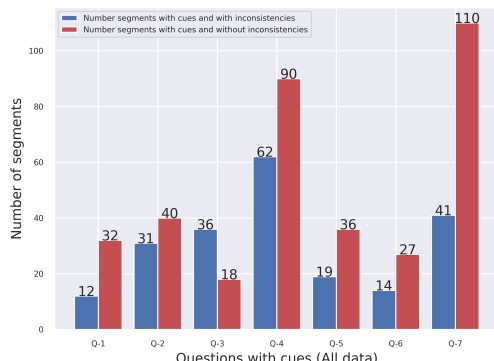

(b) Cues distribution with and without cross-modality inconsistencies over questions.

Figure 19: Distribution of cues with and without cross-modality inconsistencies over segments; in addition to question-based distribution used by annotators in BAH dataset.

## I  BAH DATASET ANNOTATION CODEBOOK

This section contains relevant information regarding our designed annotation codebook for A/H recognition. We provide the definitions of A/H and the types of cues (Table 9), as well as a more detailed description of the most relevant cues in each modality used to detect A/H, which include facial cues (Table 10), language cues (Table11), audio cues (Table 12), body language cues (Table 13), and cross-modal inconsistency cues (Table 14). The codebook is a working document that continues to evolve in response to relevant insights emerging from expert annotations and contributions from behaviour change experts. Updates on the codebook will be made available and communicated upon request. This iterative approach aligns with established qualitative research practices, where coding frameworks are refined throughout the analysis process to better reflect the complexity and richness of the data (Bradley et al., 2007).

| Term | Definition |
|---|---|
| Ambivalence/Hesitancy | The simultaneous presence of competing positive and negative feelings, ideas, thoughts, or emotions towards one same object or goal. A state in which a person has not entirely made up their mind about doing something; when they aren't fully decided on how to act (towards a behaviour or object; not necessarily the goal behaviour; excluding towards language or answering questions) |
| Facial Cues | Different motions of the muscles in the face. Facial expressions commonly occur around the mouth and eyes, including changes in a person's gaze. They can be used to assess a person's emotional state. |
| Language Cues | Includes verbal/speech-based expressions of ambivalence or hesitancy. Some common verbal expressions can include the use of 'I want to... but...', 'mmmm', among others. |
| Audio Cues | Changes in a person's non-verbal language, such as changes in tone, speed and pitch. |
| Body Cues | Non-verbal signals that include gestures, body posture and movements. Some of the cues that can be annotated as body language are hand movements, head tilts, shoulders shrugging and sighs (chest movement). |
| Cross-modal inconsistency Cues | Simultaneous incompatibility between two or more modalities or different types of cues. For example, this could be represented by someone saying 'yes' while shaking their head side to side. |

Table 9: `BAH` dataset annotation codebook: definitions.

| Facial cues | Definition |
| --- | --- |
| **Upper Region** | |
| Close 2 | A change in the frequency with which someone blinks or closing one's eyes for a longer period (e.g., either keeping them closed, or blinking for a long time). This excludes normal blinking, it is annotated when there is a difference compared to the participants baseline. Closing of both eyes; "blinking" (with both eyes). |
| Close 1 | Closing one eye at the time; includes winking. The duration of the wink is not relevant, it can be a quick wink or a longer one. |
| Squint | Partially closing one or both eyes. Significant or identifiable changes or contractions in the muscles around the outer or inner corners of the eyes. It might involve some changes in the eyebrows, forehead and cheeks. Includes squinting eyes, muscles contracting around the eyes. |
| Frown | To bring your eyebrows together (inner eyebrow) so that there are lines on your face above your eyes. Frown, forehead fold, small frown, tensed forehead, wrinkled forehead, furrowing brows, lowering inside corners of eyebrow |
| Eyebrow | Lowering/raising external parts of eyebrow(s) (or full eyebrow(s)). [i.e., one or both eyebrows] |
| Gaze | Changes in the direction of the gaze by moving the eyes. Moving eyes (not face) to look down, up, to the side. |
| Eye roll | Eye-rolling is a transitory gesture in which a person briefly turns their eyes upward, often in an arcing motion from one side to the other. The eyes do not set on anything in particular and go back to their previous position. |
| Open | Opening the eyes, looks like an increase in awareness. Eyes look slightly bigger. Engagement of the eyelids, contracting the eyelid muscles to make them look wider. |
| **Lower Region** | |
| Smile | Ends of the mouth/lips curve up, often with the lips moving apart. Includes: Smile, smirking, half a smirk, fake smile, raising both sides of the mouth, side smile, half smile |
| Pout | Pushing one's lips or one's bottom lip forward; or turning the outer sides of the lips downwards. Pouting, pursed lips, "frowning" with one's mouth |
| Lip press | Contracting or pressing lips without pushing them forward. Includes: pressed lips, pressing lips together, putting lips together. Excludes pressing lips to pout/purse. |
| Hanging | Leaving one's mouth open for an extended time (e.g., hanging mouth, gaping mouth). |
| Mouth | Any other movements of the mouth that (1) are not captured by smile, pout, or pressing lips, and (2) is not a result of the baseline speech patterns. Opening mouth, opened mouth, raising upper lip, rising one side of the mouth, taking the mouth corners back and lower them |
| Wrinkle Chin | Moving the muscles around the chin to create identifiable lines, folds, ridges or furrows in the chin. Usually seen as a contraction of the chin muscles creating creases around or on the chin. |
| Nose | Changes in the movement or looks of the nose. Includes significant movements on the nostrils, the tip of the nose, scrunching the nose, or any other muscle movement that would create a change in the nose. |

Table 10: BAH dataset annotation codebook: facial cues.

| Language cues | Definition |
|---|---|
| Filler sound | Sound made during a pause in speech signalling the person isn't done taking. Examples: "mmm", "umm", "hum", "emmm", "err", "uh", "ah" |
| Filler word | Words used that do not contain substantive content, but are used as fillers to fill in space while the person thinks (or to signal they are not done talking, or that they are about to talk): "like", "you know", "I mean", "okay", "so", "actually", "basically" |
| Hedging | Words/expressions used to express ambiguity about what one is saying (about to say or just said). Examples: "somewhat"; "I'm not an expert, but..."; "… right?"; "… isn't it?"; "I do not know…"; "all I know"; "I think…" |
| Correction | Corrects something they said. This focuses on the content of what is said, not on a syntax-based error, or speech error. |
| Repetition | Emphasizing a phrase by repeating, or repetition of a word, might be related to trying to find the right word or expression |
| Com-B Constructs | |
| Positive | Statement of positive feelings towards a behaviour or action. |
| Negative | Statement of negative feelings towards a behaviour or action. |
| Excuse | Statement where the participant mentions an excuse, a reason or justification for something that has happened or hasn't happened. It can also be an expression of regret for doing/not doing something. Use of 'but'. Shows avoidance or lack of responsibility |
| Success | Statement of success with goal (focused on the behaviour) |
| Fail | Statement of failure with goal (focused on the behaviour) |
| Cap | Mentions having the capability to change their behaviour. Includes physical capability (e.g., balance, dexterity) or psychological capability (e.g., knowledge, skills, memory). |
| No cap | Mentions NOT having the capability to change their behaviour. Includes physical capability (e.g., balance, dexterity) or psychological capability (e.g., knowledge, skills, memory). |
| Mot | Mentions having motivation to change their behaviour. Includes reflective motivation (e.g., making plans, having positive attitudes/beliefs) or automatic motivation (e.g., desires, habit, feelings) |
| No mot | Mentions NOT having motivation to change their behaviour OR motivation NOT to change their behaviour. Includes reflective motivation (e.g., making plans, having positive attitudes/beliefs) or automatic motivation (e.g., desires, habit, feelings). |
| Opp | Mentions having opportunity to change their behaviour. Includes physical opportunities (e.g., access to financial resources, location, time) or social opportunity (e.g., support/encouragement from others; norms) |
| No opp | Mentions NOT having opportunity to change their behaviour. Includes physical opportunities (e.g., access to financial resources, location, time) or social opportunity (e.g., support/encouragement from others; norms) |

Table 11: `BAH` dataset annotation codebook: language cues.

| Audio cues | Definition |
|---|---|
| Pause | Briefly interrupting a sentence by having silent pauses in between words or ideas that differ from the usual pace of how the participants speaks. It includes silent pauses, paused speech or ideas. |
| Cut words | Ending speaking a word before completing the utterance of the word (e.g., say "exer..." instead of "exercise"). Breaking the words or interrupting the words while they are being spoken. Might involve correcting syntax/speech |
| Slow | Reducing the speed of speaking. There is a perceptible change in the speed while someone is talking, making it slower or de-accelerated. It differs from paused speech or cutting off words since the words, phrases or ideas are not cut off or left in the middle, there are no significant silences in the answers. It can include elongating syllables or words. Speed change is determined in comparison to the person's own baseline. |
| Fast | Changes in the speed of the answers, making it faster. Information is given quickly, briskly or lively. Speed change is determined in comparison to the person's own baseline. |
| Volume | Changes in volume of speech. Differences in how loud or quiet an answer is shared, or there can be differences in the volume of specific words or syllables. Includes: Raising volume, lowering volume, high volume, low volume, and mumbling. Volume change is determined in comparison to the person's own baseline. |
| Shaky voice | When there is an rapid fluctuation or trembling of rhythm or tone (i.e., there is instability) to the way someone is speaking. It includes voice shaking, quivering in voice. Excludes case when shaking is due to laughing |
| Breath | Audible breath, inhaling or exhaling, it can be while the person is talking or before/after a phrase. It includes changes in the breathing rhythm, intensity or deepness of the person (compared to the baseline) that create a sound. Includes sigh, deep breath |
| Click | Quick sound made by pressing the tongue against the roof of the mouth or back of the teeth and snapping it downward. It often signals disapproval, unsureness or impatience. The sound resembles a "tsk" or "tsk-tsk." |
| Laugh | Engaging in laughter, or variations thereof (e.g., snicker, chortle, giggle) |
| Stuttering | Involuntary repetition of sounds while speaking. This can be seen as a disruption or blocking of the speech by prolongation sounds or by struggling to say a word or a part of a word. Even though the stutter might cut off a word or phrase it is different since the person will finish the word or idea. Includes stammer, stumble. |

Table 12: BAH dataset annotation codebook: audio cues.

| Body cues | Definition |
|---|---|
| Look away | Moving the orientation of the head away from the baseline position such that eyes or the gaze will look away. Includes the head facing down, head facing up, looking down, looking up, looking from side to side, lowering head, raising head. |
| Shake | Turning the head from side to side, it can be done with repetitive head movements or with a slight turn of the head to one or both sides. Includes shaking head "no". Rotation is on the horizontal plane |
| Tilt | Angling the head to the side without focusing on something else, and holding the position. Changing the position of the head so it is in a sloping position. It can be accompanied by changes in the gaze but not necessarily. Includes head tilting up and down, tilting head to the side, tilted head. Includes bobbling head. |
| Throw | Throwing the head in a rapid movement in a particular direction. |
| Sigh | Movements of the chest, shoulder or head that accompany a sigh or a deep breath. It includes long sigh, deep breath, sigh, big sigh. Noticeable bringing the chest or diaphragm muscles up and down. Change determined in comparison to the person's own baseline. |
| Nod | Moving the head up and down. Lowering and raising the head, it can be done by slight or clearly marked movements. Includes movements such as back and forward or a single small nod. |
| Shrug | Raising of the shoulders, it can be a momentary or slight rise or a longer movement where one or both shoulders is raised. It includes shrugging shoulders, shrugs |
| Hands | Movements or placement of the hands that differs from baseline |
| Posture | Movements in the overall positioning of the spine, body or arms (independent from the head). The changes are determined by each person's baseline. It includes movements like readjusting in the seat, sloughing, turning to the sides. Needs to involve more than just the head. Excludes shrugging. |
| Scratch | Movements in the hands and arms to scratch or caress another part of the body or face. It includes scratching head, scratching neck, scratching eyes, scratching chin |
| Restless | Rhythmic and repeated movements. Can be swaying, shaking, being jittery. |

Table 13: BAH dataset annotation codebook: body cues.

| Cross-modal inconsistency | Definition |
|---|---|
| FL | Face and language/speech do not match. E.g., looking uncomfortable while saying yes, looking annoyed or uncomfortable while saying they are happy, smiling while saying they are worried. |
| FA | Face and audio do not match. E.g., speaks in a sad, energetic tone while smiling. |
| FB | Face and body do not match. E.g., Nodding while looking afraid or concerned, showing disgust but leaning forward |
| LA | Language/speech and audio do not match. E.g., speaks in a sad, energetic tone while saying they are happy. |
| LB | Language/speech and body do not match. E.g., seems like they are about to say something but do not, nod is discrepant with verbal speech, shaking head while saying yes |
| AB | Body and language/speech do not match. E.g., unengaged tone while nodding (in agreement) |

Table 14: BAH dataset annotation codebook: cross-modal inconsistency cues - occurring simultaneously.

## J    BASELINE RESULTS

| Visual features | AVGF1 | AP |
|---|---|---|
| Cropped faces | **0.5028** ± 0.0078 | **0.1923** ± 0.0054 |
| Full frame (body) | 0.4781 ± 0.0209 | 0.1774 ± 0.0063 |
| Head-pose | 0.4321 ± 0.0213 | 0.1915 ± 0.0021 |

Table 15: Impact of visual modality type on model performance on test set of `BAH` at frame-level classification. ResNet18 backbone is used. We report the average and standard deviation of 5 repetitions with different seeds.

### J.1    VISUAL MODALITY: CROPPED FACES VS FULL FRAME VS HEAD-POSE FEATURES

While cropped faces are the de facto visual modality used in affective computing, other visual features could be also relevant in A/H recognition. We explore here two additional visual features that are related to cues used by annotator to detect A/H. In particular, we consider the full frame which covers the body which is an important cue. In addition, we use head-pose estimation which covers the body cue "Look away". To estimate the head pose, we use the pretrained model TokenHPE (Zhang et al., 2023a). The backbone is frozen to yield head-pose features, which are followed with an A/H classifier layer. Results obtained in Tab.15 show that while cropped faces and full frame yield competitive results, head-pose features have poor results in terms of `AVGF1`. This suggests that using only head-pose is not enough as many other cues are lost. However, it could be used to model context. Additionally, cropped faces results are more stable. We note also that the full frame case contains noisy background which may have contributed in decreasing performance. Segmenting the body alone could yield better performance. In all our experiments, we use cropped faces for the visual modality.

### J.2    SUPERVISED LEARNING CASES (FOLLOWUP FROM MAIN PAPER)

**Training details.** For both cases, pretraining visual models on `RAF-DB` (Li et al., 2017), `AffectNet` (Mollahosseini et al., 2019), and `Aff-wild2` (Kollias & Zafeiriou, 2019), their fine-tuning, and final training with multimodal setup, we used a learning rate between 0.0009 to 0.001 with multiplying coefficient of 10. When training on `BAH`, and in the case of using context, we used a window size between 24 (1 second) to 2880 (2 minutes) with a step of 1 second (24 frames). In this case, we use a mini-batch size in $\{2, 4, 8, 16, 32\}$, where a sample in the mini-batch is a window of frames; and a single-GPU training. In all trainings, we used a weight decay of 0.0001. All our experiments were conducted on a server with 4 NVIDIA A100 GPUs with 40 GB of memory, AMD EPYC 7413 24-Core Processor, and 503GB of RAM. We present in Table 16 the computation time of the multimodal case.

**Ablation over the window length.** We conduct an ablation to study the impact of the context (window length) on the performance of recognizing A/H. To this end, we use a window length from 24 to 3264 with a step of 1 second (24 frames). Figure 20 shows the obtained results. By considering `WF1`, performance improves with the increase of the context where it can reach above 0.825. On the other hand, `AP` prefers small context. Using a small context of few seconds could be a good compromise for all the metrics. Note that the average length of an A/H segment is around 4 seconds (96 frames).

**Video-level classification.** To obtain video level predictions, we resort to a simple post-processing of frame-level predictions in the main paper. In particular, we follow (De-la Torre et al., 2015), where a sliding window averages the probability of the positive class at each frame. The case where high probability in a window suggests the presence of A/H in the video. In our case, we consider a context of 1 second (24 frame). The maximum probability across all windows is considered the video probability to have the positive class. The probability of the negative class is the complement of the probability of the positive one. We then proceed to measure the same performance metrics `AVGF1`, and `AP`. Similar to the main paper, we report the performance at video-level for different cases: using visual backbone only without and with context (Table 17), multimodal (Table 18),

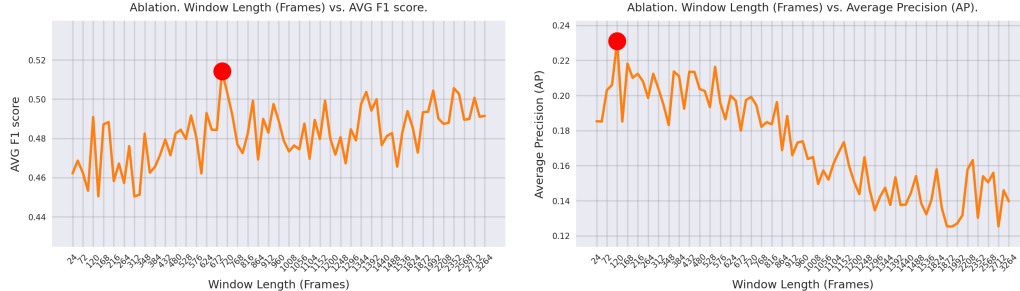

Figure 20: Impact of context (window) length on the performance of frame-level classification when using visual modality alone (ResNet18): `AVGF1`, and `AP`. Best performance is indicated in red dot.

| Case | Value |
|---|---|
| Train time 1 epoch | $\sim$ 5mins |
| Inference time per-frame | $\sim$ 0.12ms |
| Total n. params. | $\sim$ 223M |
| N. learnable params. | $\sim$ 5M |
| N. FLOPs | $\sim$ 1.87 TFLOPs |
| N. MACs | $\sim$ 938 GMACs |

Table 16: Computation time, number of parameters, number of FLOPs/MACs for multimodal case with visual, audio and text (with ResNet152 for visual backbone). Visual backbone is frozen, while audio and text backbones are used to extract features offline and store them.

| | Without context | | With context (TCN) | |
|---|---|---|---|---|
| Backbone | `AVGF1` | `AP` | `AVGF1` | `AP` |
| APViT (Xue et al., 2022) | **0.5871** | 0.2882 | **0.6134** | **0.3703** |
| ResNet18 (He et al., 2016) | 0.4445 | 0.0993 | 0.4425 | 0.1437 |
| ResNet34 (He et al., 2016) | 0.4419 | 0.1598 | 0.4448 | 0.2188 |
| ResNet50 (He et al., 2016) | 0.4405 | 0.1456 | 0.4442 | 0.1247 |
| ResNet101 (He et al., 2016) | 0.4236 | 0.1315 | 0.4448 | 0.1622 |
| ResNet152 (He et al., 2016) | 0.4715 | **0.1852** | 0.4316 | 0.2053 |
| Video-FocalNet (tiny) (Wasim et al., 2023) | – | – | 0.5294 | 0.6545 |
| Video-FocalNet (small) (Wasim et al., 2023) | – | – | 0.5333 | 0.6558 |
| Video-FocalNet (base) (Wasim et al., 2023) | – | – | 0.5663 | **0.6732** |

Table 17: Visual modality performance on test set of `BAH` at video-level classification: impact of architecture and context.

and fusion (Table 19). Similar to frame-level results, using context and multimodal yields better performance. In addition, FAN fusion yields the highest performance over both metrics.

In addition, we conducted experiments on Video-FocalNet (Wasim et al., 2023) model, specialized in video-based classification. It can be trained to directly predict a video class based on sampled frames. However, it accounts only for visual modality. We used cropped-and-aligned faces as input. We experimented with different architectures: tiny, small, and base. Pretrained weights provided by (Wasim et al., 2023) over Kinetics-400 dataset (Kay et al., 2017) are used for initialization. Following (Wasim et al., 2023), we sampled 8 frames per-video. The stride between the frames depends on the video length. This is done during training and test. We used a batch size of 64 (videos) over 4 GPUs. The model is finetuned over `BAH` dataset for 60 epochs. Table 17 shows that such specialized model yields better performance in average over both metrics. Although performance fluctuates between the three architectures, base-case yields an `AP` of 0.6732 and `AVGF1` of 0.5663, leading the board in average.

| Modalities | AVGF1 | AP |
|---|---|---|
| Visual | 0.4448 | 0.1622 |
| Audio | 0.4546 | **0.5306** |
| Text | 0.5589 | 0.4731 |
| Visual + Audio | 0.4428 | 0.1964 |
| Visual + Text | 0.5614 | 0.3407 |
| Audio + Text | 0.4366 | 0.3148 |
| Visual + Audio + Text | **0.5934** | 0.3219 |

Table 18: Multimodal models performance on test set of `BAH` at video-level classification. For visual modality, ResNet152 backbone is used.

| Fusion type | AVGF1 | AP |
|---|---|---|
| LFAN (Zhang et al., 2023b) *(cvprw,2023)* | **0.5934** | **0.3219** |
| CAN (Zhang et al., 2023b) *(cvprw,2023)* | 0.4011 | 0.3206 |
| MT (Waligora et al., 2024) *(cvprw,2024)* | 0.4448 | 0.1069 |
| JMT (Waligora et al., 2024) *(cvprw,2024)* | 0.4448 | 0.0993 |

Table 19: Feature fusion performance on test set of `BAH` at video-level classification.

### J.3 ZERO-SHOT INFERENCE: MULTIMODAL LARGE LANGUAGE MODELS (M-LLMS)

Multimodal LLMs (M-LLMs) have gained significant attention in the affective computing space due to their ability to infer cross-modal dynamics across the visual, aural, and textual modalities. The problem of detecting ambivalence and hesitancy in videos is inherently multimodal as it requires also capturing the cross-modal inconsistency. To get out-of-the-box performance of existing SOTA M-LLM, we performed zero-shot inference using the 'Video-LLaVA-7B-hf' (Lin et al., 2024). Since the performance of an M-LLM or LLMs in general can be heavily influenced by the query prompt, we experiment with different variations of the prompts. Table 20 summarizes the different prompt variations used for zero-shot inference.

| | **Prompt** |
|---|---|
| Simple | 'Classify the emotion in the video as either '*Non-Ambivalen*t' or '*Ambivalent*'.' Respond with only one word: ' |
| Definition 1 | 'Definition: Ambivalence is the state of having contradictory or conflicting feelings or attitudes towards something or someone simultaneously. Classify the emotion in the video as either '*Non-Ambivalent*' or '*Ambivalent*'. Respond with only one word: ' |
| Definition 2 | 'Definition: Ambivalence and Hesitancy is understood as the simultaneous experience of desires for change and against change. Classify the emotion in the video as either '*Non-Ambivalent*' or '*Ambivalent*'. Respond with only one word: ' |
| Transcript + Def 1 | 'Video transcript: {transcript}. Definition: Ambivalence is the state of having contradictory or conflicting feelings or attitudes towards something or someone simultaneously. Classify the emotion in the video as either '*Non-Ambivalent*' or '*Ambivalent*'. Respond with only one word: ' |
| Transcript + Def 2 | 'Video transcript: {transcript}. Definition: Ambivalence and Hesitancy understood as the simultaneous experience of desires for change and against change. Classify the emotion in the video as either '*Non-Ambivalent*' or '*Ambivalent*'. Respond with only one word:' |

Table 20: Summary of prompt variations for zero-shot inference.

### J.3.1 Frame-Level Prediction

For frame-level prediction, we adopt a segment-wise strategy, where the entire video is divided into 8-frame chunks and passed through the model using a sliding window. This way, the model sees all the frames in each video. A single prediction is obtained for the window, which is replicated for the segment to match the total number of frame labels in each video. The model's output, 'Non-Ambivalent' or 'Ambivalent', is mapped to 0 and 1 respectively to match the ground truth.

| Prompt | AVGF1 |
|---|---|
| Simple | 0.4416 |
| Definition Only 1 | 0.4456 |
| Definition Only 2 | 0.1651 |
| Transcript + Def 1 | **0.4535** |
| Transcript + Def 2 | 0.1849 |

Table 21: Frame Level Prediction using M-LLM.

Table 21 shows the results obtained for frame-level predictions using different prompts. The best results for frame-level prediction are obtained using the 'Transcript + Def 1' prompt, where the actual transcript of the video is also provided, along with a straightforward definition of A/H. The model performs better with a more straightforward definition of the concept of ambivalence, with Definition 1.

In the segment-wise approach applied with a sliding window of 8 frames, the model essentially sees every frame, but this approach limits the context window to be 1/3 of a second, which may not be enough to capture the temporal dependencies in the visual modality. We investigate the effect of various lengths context windows on the overall performance.

| Context Window | AVGF1 |
|---|---|
| 24 Frames | 0.4539 |
| 48 Frames | 0.4542 |
| 80 Frames | 0.4535 |
| 120 Frames | **0.4546** |
| 192 Frames | 0.4540 |

Table 22: Performance comparison with increasing size context window for frame-level prediction.

We selected the best-performing query prompt from the first experiment (Table 21) to perform the ablation on the context window size. Table 22 shows the results with different lengths of context window for the visual modality. 24 frames represent a one-second context window. Increasing the context window size to 120 frames (5 seconds) only marginally improves the overall performance of the model, and it plateaus at 120 frames and then starts to drop which is an indicator that the visual encoder starts losing information with a longer context window.

### J.3.2 Video-Level Prediction

For video-level prediction, the entire video is fed to the model, and the transcript is embedded in the prompt. The model selects 8 uniformly spaced frames from the video and predicts a single output. Similar to frame-level predictions, the model's output is mapped to 0 and 1, and the performance metrics are calculated.

Table 23 presents the video-level prediction results. Similar to frame-level predictions, the 'simple' prompt without any context on the definition or the transcript performs the worst and predicts all samples to be 'Non-Ambivalent'. A similar trend is also observed here, i.e., adding the definition and the transcript substantially affects the model performance.

| Prompt | AVGF1 |
|---|---|
| Simple | 0.2827 |
| Definition Only 1 | 0.3326 |
| Definition Only 2 | 0.3772 |
| Transcript + Def 1 | **0.6341** |
| Transcript + Def 2 | 0.3945 |

Table 23: Video Level Prediction using M-LLM.

### J.3.3 ANALYSIS AND DISCUSSION

The performance of M-LLM with zero-shot inference is substantially influenced by the query prompt. As observed from tables 21 and 23, simply asking the model to predict emotion based on the visual modality only performs the worst, whereas adding only the definition of A/H in the query prompt helps the model better identify the positive(A/H) class. Best results in all cases are obtained with the introduction of the text transcript of the video in the query prompt. We conjecture that this happens for two reasons: i) the textual modality serves a significant role in the identification of the A/H class, and ii) the current M-LLMs' performance is heavily reliant on the textual modality. This coheres with the overall structure of traditional M-LLMs that are built upon well-trained LLMs with the addition of a visual encoder like ViT, which is used to encode the visual information that is fed to the LLM for downstream tasks. Intuitively, the performance should increase with careful fine-tuning on the BAH dataset.

Further, the idea of textualizing the aural and visual modalities explored in (Richet et al., 2024) can be well-suited for a task like this where the audio and visual modalities essentially summarize the cues detected in the corresponding modalities. Particularly for tasks like subtle emotion recognition or the detection of A/H, where cross-modal inconsistency has to be considered. Textualizing the aural and visual modalities can be done to adequately exploit the reasoning abilities of SOTA LLMs.

### J.4 PERSONALIZATION USING DOMAIN ADAPTATION

Domain adaptation (DA) (Han et al., 2020; Li & Deng, 2018) has emerged as a promising approach for personalized expression recognition, where the model is trained on diverse labeled source data to generalize to unlabeled target domains representing individual users. Recent research emphasizes on subject-based domain adaptation (Sharafi et al., 2026; 2025; Zeeshan et al., 2024; 2025), where each individual is defined as a distinct domain. DA will be employed to personalize ML models by considering each participant in the test set as a separate target domain.

**Experimental Protocol.** For personalized in BAH, we adopt the standard protocol from prior work (Zeeshan et al., 2024; Sharafi et al., 2025), which involves partitioning the data of each target individual into train, validation, and test sets. Given the class imbalance in the BAH dataset, we ensure a balanced representation of positive and negative samples within each split. We establish the following baseline methods to evaluate the effectiveness of personalized BAH recognition: **Source-only**: The model trained on the source data is directly evaluated on the target individual test set without any adaptation. This assesses the generalization capability of the source model. **Unsupervised Domain Adaptation (UDA)**: Source data with labels is utilized to adapt the model to each target individual using unlabeled data. This explores the potential of leveraging source knowledge for personalization in the absence of target labels. **Source Free Unsupervised Domain Adaptation (SFUDA)**: Adaptation is performed solely using the unlabeled data from the target individual, without access to the source data. This examines the feasibility of personalization when the source data is unavailable. **Oracle**: The model is fine-tuned using the labeled data from the target individual during training. This provides an upper-bound performance, representing a fully supervised model.

**Visual backbone.** We employ a ViT-based model for personalization, leveraging its superior performance over ResNet-based architectures for visual tasks without contextual information. In all our experiments, we utilize a ViT-based model pre-trained on the source data.

| Methods | AVGF1 | AP |
|---|---|---|
| Source-only | $0.4894 \pm 0.0999$ | $0.3565 \pm 0.1841$ |
| UDA (MMD) (Sejdinovic et al., 2013) | $0.4931 \pm 0.0943$ | $0.3589 \pm 0.1831$ |
| UDA (Sub-Based) (Zeeshan et al., 2024) *(fg,2024)* | $\mathbf{0.5417 \pm 0.0728}$ | $\mathbf{0.3739 \pm 0.1789}$ |
| SFUDA (SHOT) (Liang et al., 2020) *(icml,2020)* | $0.4919 \pm 0.1056$ | $0.3520 \pm 0.1656$ |
| SFUDA (NRC) (Yang et al., 2021) *(neurips,2021)* | $0.5174 \pm 0.1041$ | $0.3688 \pm 0.1487$ |
| Oracle | $0.5864 \pm 0.0751$ | $0.4181 \pm 0.1750$ |

Table 24: Performance of UDA and SFUDA with Source-only and Oracle on BAH. Results are reported as the average over all target subjects (mean $\pm$ standard deviation).

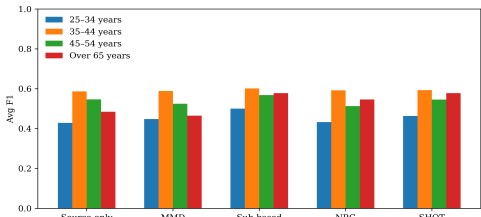
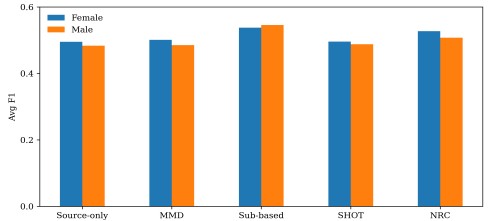

(a) Comparison between different age groups on DA methods.

(b) Comparison between different DA methods based on participant Sex.

Figure 21: Comparison between different age groups and sex on DA methods.

### J.4.1 UNSUPERVISED DOMAIN ADAPTATION:

We investigated two unsupervised domain adaptation (UDA) approaches for personalized BAH recognition: (i) a discrepancy-based method using Maximum Mean Discrepancy (MMD) (Sejdinovic et al., 2013) to minimize the domain gap and improve performance on the target subject, and (ii) a subject-based (Zeeshan et al., 2024) method using self-supervision that trains the model by generating pseudo-labels for the target domain, followed by reducing the domain shift using MMD that aligns source and target.

**Implementation detail.** In UDA experiments, we optimize our model using Stochastic Gradient Descent (SGD) (Sutskever et al., 2013) with a learning rate of $2e - 4$, momentum of $0.9$, weight decay of $5e - 4$, and a cosine annealing scheduler (Loshchilov & Hutter, 2017) with a minimum learning rate of $2e - 5$. We set the batch size to $64$ and run each target adaptation for 10 epochs. For the subject-based method, we introduce a hyperparameter $\gamma_3 = 0.01$ to weight the target loss, computed using pseudo-labels generated by the Augmented Confident Pseudo-Label (ACPL) technique (Zeeshan et al., 2024). This weighting is essential for mitigating noise in the pseudo-labels, in conjunction with a confidence threshold of $0.95$ that is updated every 4 epochs.

### J.4.2 SOURCE FREE UNSUPERVISED DOMAIN ADAPTATION

Two source-free unsupervised domain adaptation (SFUDA) approaches were explored for personalized BAH recognition: (i) a representation learning strategy inspired by hypothesis transfer (Liang et al., 2020), where information maximization was used to adapt the model to the target domain, and target-specific prototypes guided pseudo-labelling for class-level alignment, and (ii) a neighbourhood-based method (Yang et al., 2021) in which label consistency was encouraged among target features and their reciprocal nearest neighbours, while expanded neighbourhoods were used to aggregate local structure and reduce the impact of noisy supervision through self-regularization and affinity-weighted loss.

**Experimental protocol.** We optimize the model using SGD with a learning rate of $1e - 4$, momentum of $0.9$, and weight decay of $1e - 3$. The model is trained for 30 epochs with a batch size of 64. For NRC-based adaptation, we maintain memory banks of target features and predictions to retrieve $K = 3$ nearest neighbours and $M = 2$ expanded neighbors.

### J.4.3 RESULT AND ANALYSIS

The average performance across target participants, along with the standard deviation, is presented in Table 24. All reported results are based on evaluations of the respective target test sets. Our analysis demonstrates the effectiveness of domain adaptation for personalized detection of the A/H class in the BAH dataset. All tested methods surpass the Source-only baseline in `AVGF1` and `AP` positive-class metrics. Notably, Sub-based achieves the highest `AVGF1` (0.5417) and `AP` (0.3739), outperforming other domain adaptation techniques. While MMD and SHOT yield only modest gains, they still lag behind the Sub-based method, underscoring the advantage of pseudo-labeling for boosting minority-class recall and precision. In contrast, NRC achieves a more substantial improvement of around 3 percentage points over Source-only, indicating that even with some sensitivity to domain shifts, it more effectively exploits target-domain structure. The Oracle upper bound (`AVGF1`: 0.5864) underscores the substantial potential for further advancements in positive-class detection within this context. Even slight degradations in negative-class performance disproportionately impact the overall `AP` score. For example, Sub-based method emphasizes enhancing positive class identification, likely incurs a cost in precision or recall on the more frequent negative class, a necessary compromise to effectively detect the A/H class.

**Age-wise analysis.** Figure 21a illustrates the varying impact of age on DA methods. Overall performance tends to peak in the *35–44 years* group across most methods, while the *25–34 years* group generally shows the lowest `AVGF1`. **Sub-based** consistently achieves the highest or near-highest performance in all age groups, with clear gains over Source-only, and is only slightly outperformed by **SHOT** in the *Over 65 years* group. **SHOT** remains competitive across all ages and, together with Sub-based, yields the largest improvements for the youngest (*25–34 years*) and oldest (*Over 65 years*) subjects. In contrast, **MMD** and **NRC** provide only modest gains over Source-only in the *25–34 years* and *35–44 years* groups and can even underperform Source-only for *45–54 years*, highlighting their sensitivity to age-related domain shifts. Finally, **Source-only** has consistently lower Avg F1 across all age groups, confirming the benefit of DA, particularly for the more challenging age ranges.

**Sex-wise analysis.** In the Figure 21b, we can observe that the *female* category generally exhibits higher values across most methods compared to the *male*. Specifically, *female* shows the highest values in **NRC** (0.0.52), **SHOT** (0.0.49), and **MMD** (0.50). It can also be noted that the number of female subjects is equal to the male subjects.

