# OpenReview forum: "BAH Dataset for Ambivalence/Hesitancy Recognition in Videos for  Digital  Behavioural Change"
_ICLR.cc/2026/Conference — ICLR 2026 Poster_

### Official Review · Reviewer_pr3G · 2025-10-24

**Soundness:** 3
**Presentation:** 3
**Contribution:** 3
**Rating:** 6
**Confidence:** 4

**Summary:**

The paper introduces a new multimodal dataset for recognizing ambivalence and hesitancy. It includes recordings with expert annotations covering facial, vocal, verbal, and bodily cues, offering a valuable resource for studying complex emotional states relevant to behavior change. The authors provide baseline experiments across different modalities and fusion strategies, showing that temporal context and multimodal learning improve performance but that the task remains challenging, paving the way for future research on nuanced emotion understanding.

**Strengths:**

1. The first public dataset focused on recognizing ambivalence and hesitancy, emotional states that play an important role in behavior-change research but have received little attention in machine learning.
2.The authors conducted comprehensive experiments comparing different setups: single-modality vs. multi-modality, with or without temporal context, and various fusion methods. The results show that adding temporal context improves performance, simple concatenation often works surprisingly well, and combining all three modalities does not always outperform simpler setups, indicating that multimodal fusion for ambivalence recognition remains a challenging problem.

**Weaknesses:**

1. The dataset provides only binary (0/1) labels for Ambivalence/Hesitancy, which may be too simplistic. Incorporating continuous annotations such as valence–arousal or PAD scales could enable a more fine-grained analysis of emotional states and cues.
2. Although the dataset includes facial, linguistic, audio, and body cues, these are annotated only at the video level. While this adds interpretability, it limits temporal precision. Adding timestamps or ideally frame-level cue annotations would make the dataset far more useful for detailed temporal modeling.
3. The paper describes the annotation process and co-annotation procedures but does not report quantitative inter-annotator agreement. Without this metric, it is difficult to assess the reliability and consistency of the labels.
4. The paper does not provide metadata statistics for the training, validation, and test splits, particularly regarding demographic balance. Reporting and ensuring demographic diversity across splits would support fairness analysis and enhance the dataset’s research value.

**Questions:**

1. What are the inter-annotator agreement (IAA) scores at both the video and frame levels for Ambivalence/Hesitancy labeling, as well as for cue tagging? Additionally, how do the models perform on high-certainty versus low-certainty segments?
2. How did the annotators distinguish ambivalence from mixed emotions or uncertainty without intent conflict? Please provide annotated examples and an error typology to clarify how these cases were handled.
3. Please include metadata statistics for the training, validation, and test splits, especially regarding demographic distribution, to assess data balance and fairness.
4. It would strengthen the paper to clearly explain the conceptual and methodological differences between Ambivalence/Hesitancy recognition and temporal emotion recognition, highlighting why A/H detection requires distinct modeling strategies.

---

> ### Author Response · Authors · 2025-11-21
> **Rebuttal**
>
> We thank the Reviewer for the constructive review, and for acknowledging our contributions, including the novelty of the multimodal dataset and the comprehensive experiments. Similar positive feedback is shared by the other three Reviewers. We address each weakness and question mentioned below, and incorporate all feedback in the manuscript. “W” indicates weakness, “Q” indicates question, while “A” is our answer.
>
> Since the submission of this work, we have collected and annotated an additional 76 participants with a total of 309 videos. The final BAH dataset will have videos from 300 participants. We request permission from the Reviewer to update the dataset. We are working to reproduce all the results with this new version and update the manuscript before the end of the discussion period. In the current updated manuscript, we included the following changes (in red):
> * The link to the new BAH dataset with 300 participants: https://142.137.245.13/index.php/s/MyY2GyzBwjNXFLq (under folder: BAH_DB_300_ICLR26_REVIEW) , please use the same password provided in the initial submission to unzip the file:
> @ICRL_oSY5QhGTHH5ckAf3qKCF_2026_Brazil
>
>     Authors’ identity and relevant information have been redacted. Please do not select with mouse regions in black in the documents.
>     For the first time usage of the link, please accept the security certificate on your internet browser before proceeding. The website and the zip file are safe.
> * Updated all statistics and figures about the update size of the dataset (300 participants).
> * Introduced a new section that covers recommendations for future works to better design methods for A/H recognition (Sec.4)
> * Included new experiments using different ways of visual modality instead of cropped faces. This covers: full frame, and head-pose features.
> * Included analysis of annotator cues (Sec.H).
> * Included inter-annotator agreement analysis (Sec.2.1).
> * Included demographic statistics of train, validation, and test sets (Sec.G)
> * Included distribution of participants birth country (Sec.F)
> * The published BAH dataset is now shipped with the following additional information:
>     * Annotator identifier (ID)
>     * Subject meta-data including:
>         * Age of the participant.
>         * Age range of the participant.
>         * Birth country of the participant.
>         * Canada province where the participant lives.
>         * Ethnicity of the participant.
>         * Ethnicity simplified: a simplified version of the participant ethnicity.
>         * Sex of the participant.
>         * Is participant a student or not.
>         * Use in publications: can the videos and the data of the participant be used or not for publications.
>         * Use in challenges: can the videos and the data of the participant be used or not for challenges.
>     * Code to extract frames
>
> **W-1: Continuous scale for A/H.**
>
> **A**:  Thank you for this comment. Incorporating a continuous annotation for A/H implies producing levels of intensity. While this concept is applicable to phenomena such as basic emotions or pain, where intensity often builds and then subsides,  A/H typically does not follow such a trajectory. Instead, ambivalence and hesitancy are characterised by a more sustained, subtle internal conflict between competing motivations or feelings. As such, there is generally no identifiable apex or peak moment during an A/H episode that suggests the notion of intensity. For this reason, we chose to annotate A/H as a temporally bounded event using start and end times, which aligns more closely with how A/H manifests and supports its temporal localisation in applied settings. We agree that exploring finer/continuous temporal dynamics could be valuable in future work, but it would require developing new annotation approaches suited to the specific nature of A/H. We have clarified this in the main paper (Sec. 2.1).

---

> > ### Author Response · Authors · 2025-11-21
> > **next response**
> >
> > **W-2: Clarify annotation timestamp.**
> >
> > **A**: Thank you for this comment. The annotator provides the used modalities to determine A/H and also the cues at temporal level and not at video level. Each segment of A/H has its timestamp (start and end time) allowing for better precision. Additionally, several non-overlapping segments could occur within a video (roughly 195 videos). We apologize for the confusion, and have clarified this aspect in the main paper (Sec. 2.1).
> >
> > **W-3/Q-1: Clarification about the multi-annotation.**
> >
> > **A**: Thank you for this comment. Three annotators labeled the data. To ensure consistency and clarity, all annotators underwent structured training using a standardized protocol, supported by a cue list and annotation guidelines (codebook). Weekly meetings were held to discuss challenging or ambiguous cases, and collaborative review was encouraged during the training phase and when needed throughout annotation. This process helped align annotator interpretations and supported the development of shared understanding of the constructs. Only very few cases required this collective decision. The three annotators split the data participant-wise among themselves for fast annotation where each video was labeled by one annotator. Note that annotating a single video is a resource intensive process. The decision to divide the dataset among annotators was made to allow us to annotate a larger number of participants and ensure the dataset would be of sufficient size and diversity to support downstream machine learning applications. Our priority was to strike a balance between quality and scalability. We clarified this aspect in the main paper (see Sec. 2.1), and apologize for this confusion.
> >
> > To ensure the quality of the annotation, we perform a multi-annotation of a random subset of 10 videos to assess inter-annotator agreement. We considered Fleiss's kappa measure (REF1) At the global level, the score measure was 0.65 (substantial agreement) while at the frame level, it was 0.41 (moderate agreement). When considering only videos where the annotators agreed on the presence of A/H, the frame level score increased to 0.50. This evaluation is now included in the main paper (see Sec. 2.1).
> >
> > As for high- versus low-certainty segments, we do not provide performance analyses based on that distinction. The certitude values in our annotations are not intended as labels of data quality or as categories for model evaluation; rather, they document the annotator’s subjective confidence when identifying the presence of A/H in a given segment. This information is included to promote transparency in the annotation process and to reflect the inherent ambiguity that characterizes A/H expressions.
> >
> > [REF1]: Fleiss, J. L. (1971). Measuring nominal scale agreement among multiple raters. Psychological bulletin, 76(5), 378. https://doi.org/10.1037/h0031619

---

> ### Author Response · Authors · 2025-11-21
> **next response**
>
> **W-4/Q-3: Meta-data of splits.**
>
> **A**: Thank you for this comment. We provided the meta-data of all splits in the supplementary materials (see Sec. G) in the updated manuscript. We have ensured that all splits follow a similar distribution as the full data.
>
> **Q-3: A/H and mixed emotions.**
>
> **A**: Thanks for this insightful question. In our framework, A/H encompass both mixed emotions (e.g., simultaneously expressing positive and negative feelings toward a behaviour) and uncertainty (e.g., being unsure, doubtful, or conflicted about engaging in a behaviour). These manifestations are consistent with the behavioural science literature, which conceptualises A/H as a family of states involving simultaneous, non-aligned orientations toward an action, whether emotional, cognitive, or motivational in nature. Because mixed emotions and uncertainty are considered expressions of A/H rather than separate constructs, annotators were not required to distinguish them from A/H, they were instructed to treat them as part of the A/H category when they appeared in relation to the behaviour under discussion.
>
>
> **Q-4: How is A/H recognition different from standard action recognition?**
>
> **A**: Thank you for this critical question. Compared to activity recognition (where different parts of the body and different object can be involved), in A/H recognition always involves  the human body, and the differences of expression can be minimal and therefore difficult to infer. Compared to standard action recognition and video-based emotion recognition task, A/H recognition is even more challenging because the expressions are more subtle, vary considerably across different individuals, and are often associated with discord according to different modalities. Given these challenges, video-based A/H recognition must rely on multiple audio, visual and textual modalities, like cropped faces, vocal tone and transcripts.
>
>
>
> We included a new analysis of annotator cues in Sec. H. The results show that:
>
> 1- All cues (facial, language, body, and audio) participate at almost the same rate, showing conflict between 28.4% and 22.7% of the time; and
>
> 2- Their co-occurrence (facial, language, body, and audio) dominates with 54.1% followed by facial-body-language cues combination with 14.7%. Such critical analysis could be used a-priori for training and constraining a model to explicitly detect conflict before making the final A/H decision.
>
> In a new Section 4, we include several recommendations that we believe could help develop more powerful A/H recognition methods. The idea of using sentiment analysis at a first level is introduced. Each modality is trained separately for sentiment analysis using off-the-shelf models. The response of these models could be used in a second level to predict A/H. We also included modeling conflict using statistical priors from our cues analysis. Such a modular and interpretable framework is expected to be more efficient than generic multimodal models. Additionally, constraints on conflicts within and cross modalities can be easily introduced, something that is not directly feasible for standard models.
>
> **Ethics concerns:**
>
> **A**: Thank you for raising this flag. However, it is not clear what bias or security issue is being indicated here. We would be glad to help answer your ethics questions. Please note that the dataset is stored on our university secured server. Access detail is provided to approved users after being vetted to limit misuse. The data file is protected with a strong password. This strategy is common to share and protect datasets.

---

### Official Review · Reviewer_GdzA · 2025-10-28

**Soundness:** 2
**Presentation:** 3
**Contribution:** 3
**Rating:** 6
**Confidence:** 3

**Summary:**

This paper presents the Behavioural Ambivalence/Hesitancy (BAH) dataset, a multimodal video corpus for recognizing ambivalence/hesitancy (A/H) states. It includes 1,118 videos from 224 participants across Canada, annotated at video- and frame-levels with onset–offset segments and multimodal cues (face, audio, text). The dataset aims to model complex, sustained emotional states rather than discrete peaks. Baseline experiments perform binary A/H vs. non-A/H classification, showing the task’s difficulty and establishing a benchmark for future affective computing research.

**Strengths:**

1. This paper introduces a novel multimodal dataset that integrates visual, audio, and textual modalities, focusing on the recognition of ambivalence and hesitancy (A/H), which are complex, sustained emotional states rather than discrete basic emotions. It thereby proposes a new affect recognition task that broadens the traditional emotion recognition paradigm.
2. The dataset is of moderate scale, containing videos from 224 participants across 9 Canadian provinces, ensuring demographic and ethnic diversity and supporting fairer modelling of human affect. This makes it a valuable contribution to the affective computing community.
3. The baseline experiments are well designed and relatively comprehensive, covering frame-level and video-level recognition in both unimodal and multimodal settings. The paper also reports zero-shot prediction results and unsupervised domain adaptation models for personalization, which further enhance the dataset’s utility and research relevance.

**Weaknesses:**

1. While the dataset itself is valuable, the title and task definition are misleading. The term “Ambivalence/Hesitancy Recognition” suggests separate classification of ambivalence and hesitancy, yet all experiments address a binary detection task (A/H vs. non-A/H) without distinguishing the two.
2. The inclusion of “for Behavioural Change” in the title appears overstated, as the study does not involve any behavioural intervention, longitudinal tracking, or pre– post change analysis. The link between recognizing A/H emotions and actual behavioural change is discussed only conceptually, not empirically demonstrated.
3. The scientific relevance of detecting A/H remains insufficiently justified. While ambivalence and hesitancy are theoretically relevant to behaviour regulation, the paper does not clarify how A/H recognition could inform or improve behavioural interventions. Strengthening this conceptual bridge would significantly enhance the contribution.

**Questions:**

1. As mentioned in weaknesses. We suggest distinct recognition of ambivalence and
hesitancy and a connection to behavioural change, while the paper only conducts a binary A/H vs. non-A/H classification without behavioural intervention. A clearer alignment between the task scope and the title would improve conceptual precision.
2. The abstract could be more concise and focused on the dataset’s conceptual contribution and key findings rather than listing detailed statistics. Streamlining this section would make the paper’s main message clearer.
3. In the Introduction, the motivation for detecting A/H remains vague. The paper would benefit from a clearer explanation of why distinguishing A/H from non-A/H emotions is important and in which practical contexts this task has value. At present, the classification objective seems somewhat detached from real-world applications. The connection to behavioural change also remains conceptual. Clarifying how A/H recognition could be used in adaptive feedback or intervention systems would make this link more convincing.
4. Finally, the explanation of the multimodal fusion results (Table 5 / Table 15) is not entirely satisfactory. The paper attributes the drop in performance with tri-modal fusion (visual + audio + text) to “modality conflicts.” However, given that ambivalence and hesitancy are themselves conflictive emotional states, such cross- modal inconsistency might in fact be an informative signal rather than noise. It is unclear why three-modality fusion causes conflict while two-modality setups do not, especially since the text modality is derived from speech. Further clarification and analysis of this phenomenon would strengthen the interpretation of the results.

---

> ### Author Response · Authors · 2025-11-21
> **Rebuttal**
>
> Thank you for the constructive review, and for acknowledging our contributions, including the novelty of the multimodal dataset, diversity of data and the comprehensive experiments. Similar positive feedback is shared by the other three reviewers. We address each weakness and question mentioned below, and incorporate all feedback in the manuscript. “W” indicates weakness, “Q” indicates question, while “A” is our answer.
>
> Since the submission of this work, we have collected and annotated an additional 76 participants with a total of 309 videos. The final BAH dataset will have videos from 300 participants. We request permission from the Reviewer to update the dataset. We are working to reproduce all the results with this new version and update the manuscript before the end of the discussion period. In the current updated manuscript, we included the following changes (in red):
> * The link to the new BAH dataset with 300 participants: https://142.137.245.13/index.php/s/MyY2GyzBwjNXFLq (under folder: BAH_DB_300_ICLR26_REVIEW) , please use the same password provided in the initial submission to unzip the file:
> @ICRL_oSY5QhGTHH5ckAf3qKCF_2026_Brazil
>
>     Authors’ identity and relevant information have been redacted. Please do not select with mouse regions in black in the documents.
>     For the first time usage of the link, please accept the security certificate on your internet browser before proceeding. The website and the zip file are safe.
> * Updated all statistics and figures about the update size of the dataset (300 participants).
> * Introduced a new section that covers recommendations for future works to better design methods for A/H recognition (Sec.4)
> * Included new experiments using different ways of visual modality instead of cropped faces. This covers: full frame, and head-pose features.
> * Included analysis of annotator cues (Sec.H).
> * Included inter-annotator agreement analysis (Sec.2.1).
> * Included demographic statistics of train, validation, and test sets (Sec.G)
> * Included distribution of participants birth country (Sec.F)
> * The published BAH dataset is now shipped with the following additional information:
>     * Annotator identifier (ID)
>     * Subject meta-data including:
>         * Age of the participant.
>         * Age range of the participant.
>         * Birth country of the participant.
>         * Canada province where the participant lives.
>         * Ethnicity of the participant.
>         * Ethnicity simplified: a simplified version of the participant ethnicity.
>         * Sex of the participant.
>         * Is participant a student or not.
>         * Use in publications: can the videos and the data of the participant be used or not for publications.
>         * Use in challenges: can the videos and the data of the participant be used or not for challenges.
>     * Code to extract frames

---

> > ### Author Response · Authors · 2025-11-21
> > **next response**
> >
> > **W-1-2-3, Q-1-2-3- Why ambivalence/hesitancy? Why behavioural change? Justify using A/H.**
> >
> >
> > **A**: Thank you for these comments.
> > This work is motivated by a real and pressing behavioural science challenge: A/H is the primary reason why individuals delay, avoid, or abandon health behaviour changes (REF1, REF2). While such cues are often detected and addressed during in-person interactions, there is currently no equivalent mechanism in digital health settings. Our long-term goal is to close that gap. The BAH dataset serves as a foundation to train models capable of recognising A/H reliably. Though the current dataset is not captured during the process of behaviour change, the cues that are provided are explicitly constructed to elicit feelings of ambivalence that are specifically related to behaviour change. As a next step, we will apply these models to data collected from actual digital health behaviour change interventions. This will allow us to test generalisability in real-world use and iteratively refine the models. Ultimately, the goal is to embed these models into digital interventions that adapt in real time to detect A/H cues. This can then be used to deliver intervention components that help to resolve ambivalence and hesitancy (REF3). Although the primary motivation stems from digital health interventions, identifying ambivalence and hesitancy has broader relevance, for example, in medical decision-making (e.g., uncertainty about treatment options), therapeutic settings where clients express mixed feelings about change, or human–AI interaction scenarios where users often hold conflicting attitudes toward automated systems. These examples illustrate that detecting A/H has utility across multiple applied contexts, not only behaviour change.
> >
> > Our dataset was developed with real-world digital health applications in mind, particularly for use in personalised behaviour change interventions where users interact with an avatar that prompts them with predefined questions. To mirror this setting, participants in our study responded to structured but genuine questions about behaviours they engage in or avoid, based on their actual experiences. This setup encourages authentic, spontaneous expression of ambivalence and hesitancy (A/H), while still maintaining consistency across participants. The BAH dataset was intentionally designed to maximize ecological validity. Participants had control over their environment and delivery, which contributed to naturalistic responses that reflect how people express A/H in real life. We clarified this point in the abstract and introduction.
> >
> > Regarding the distinction between ambivalence and hesitancy, the behavioural science literature frequently treats these constructs as closely related, as both describe an intermediate state between positive and negative orientations, or between acceptance and refusal to do something, that often constitutes a barrier to initiating behaviour change and a trigger for discontinuing interventions or change efforts. Because they manifest similarly in practice, our task is framed as a binary detection problem (A/H vs. non-A/H). Retaining both terms in the title and task description is therefore appropriate, as it reflects the underlying theory and supports the dataset’s generalizability.
> > We are in the process of revising the introduction and abstract to further clarify these points. They will be ready before the end of the discussion period.
> >
> > [REF1]: Williams, B. (2024). Working with Ambivalence – The Key to Making Change. In Practical Human Behaviour Change for the Health and Welfare of Animals, B. Williams (Ed.). https://doi.org/10.1002/9781394178889.ch8
> >
> > [REF2] Marije J. Van Gent, Marleen C. Onwezen, Reint Jan Renes, Michel Handgraaf, Betwixt and between: A systematic review on the role of ambivalence in environmental behaviours, Journal of Environmental Psychology, 2024, vol. 97.
> >
> > [REF3] Voisard B, Dragomir AI, Boucher VG, Szczepanik G, Bacon SL, Lavoie KL. Training physicians in motivational communication: An integrated knowledge transfer study protocol. Health Psychol. 2024 Nov;43(11):842-852. doi: 10.1037/hea0001395.

---

> ### Author Response · Authors · 2025-11-21
> **next response**
>
> **Q-4: Limited performance over multimodal case.**
>
> **A**: Thank you for this comment. Indeed, multi-modal and contextual modeling have only slight benefit compared to single visual modality despite using state-of-the-art multimodal and fusion techniques that topped a recent affective computing challenge (8th Workshop and Competition on Affective & Behavior Analysis in-the-Wild, ABAW). In addition, combining the 3 modalities showed a decline in AP and slight drop in AVGF1, compared to the dual modality cases. Adding more modalities is often expected to improve accuracy in standard classification problems, assuming that the modalities reflect some diversity. However, it is not the case here. We believe that developing better and adapted fusion techniques will improve the performance. The role of fusion in our task is not simply to combine modalities. It should also assess affect in each modality, then determine if there are conflicting emotions, an important indicator of AH. The main goal of our experiments is to assess the performance of standard multimodal deep learning  techniques on the A/H recognition task, and highlight their limitations. Results of our experiments  serve as baselines for future works.

---

> > ### Author Response · Authors · 2025-12-04
> > **update**
> >
> > Thank you again for your insightful comments. Following your recommendations (W-1-2-3, Q-1-2-3), we have updated the abstract and introduction to focus more on the newly introduced task, and the relation between ambivalence/hesitancy and digital behavioural change.
> >
> > Following your same comment, we have also slightly adjusted the title to “BAH Dataset for Ambivalence/Hesitancy Recognition in Videos for Digital Behavioural Change” to highlight the relation to digital health applications.

---

### Official Review · Reviewer_RBNh · 2025-11-01

**Soundness:** 3
**Presentation:** 3
**Contribution:** 2
**Rating:** 4
**Confidence:** 3

**Summary:**

This paper introduces the first Behavioural Ambivalence//Hesitancy (BAH) dataset collected for subject-based multimodal recognition of A/H in videos. It contains videos from 224 participants captured across nine provinces in Canada, with different age, and ethnicity. BAH contains 1,118 videos for a total duration of 8.26 hours with 1.5 hours of A/H. The paper also provides preliminary benchmarking results using baseline models trained on BAH for frame- and video-level recognition with mono- and multi-modal setups. It also includes results on models for zero-shot prediction, and for personalization using unsupervised domain adaptation. The limited performance of baseline models highlights the challenges of recognizing A/H in real-world videos.

**Strengths:**

1. It is interesting to have a dataset for ambivalence and hesitancy (A/H)  recognition which  involve subtle and conflicting emotions that are manifested by a discord between multiple modalities, such as facial and vocal expressions, and body language.

2. The authors conducted extensive experiments to demonstrate the potential of this dataset for frame- and video-based emotion recognition.

**Weaknesses:**

1. In the experimental results, the authors only consider CNN- and ViT-based models which are originally designed for image classification. It would be more interesting to consider video-based models to demonstrate the difficulty of the proposed benchmark.

2. While a dedicated dataset for ambivalence and hesitancy (A/H)  recognition is interesting, it is not clear what is the major difference between this task compared with other video recognition task such as activity recognition. In other words, what aspect makes this task challenging or difficult.

3. The dataset size is relatively small with only a total duration of 8.26 hours which can possibly limit the usefulness of the proposed dataset.

**Questions:**

1. In table 5, it is shown that with Visual + Audio + Text, the results are worse than other cases, can the authors explain the possible reason?

2. What are main challenges of ambivalence and hesitancy (A/H)  recognition compared with other types of emotion recognition such as anger or happy?

---

> ### Author Response · Authors · 2025-11-21
> **Rebuttal**
>
> Thank you for the constructive review, and for acknowledging our contributions, including the novelty of the dataset and the extensive experiments to demonstrate the potential of this dataset for frame- and video-based emotion recognition. Similar positive feedback is shared by the other reviewers. We address each weakness and question mentioned below, and incorporate all feedback in the manuscript. “W” indicates weakness, “Q” indicates question, while “A” is our answer.
>
>
> Since the submission of this work, we have collected and annotated an additional 76 participants with a total of 309 videos. The final BAH dataset will have videos from 300 participants. We request permission from the Reviewer to update the dataset. We are working to reproduce all the results with this new version and update the manuscript before the end of the discussion period. In the current updated manuscript, we included the following changes (in red):
> * The link to the new BAH dataset with 300 participants: https://142.137.245.13/index.php/s/MyY2GyzBwjNXFLq (under folder: BAH_DB_300_ICLR26_REVIEW) , please use the same password provided in the initial submission to unzip the file:
> @ICRL_oSY5QhGTHH5ckAf3qKCF_2026_Brazil
>
>     Authors’ identity and relevant information have been redacted. Please do not select with mouse regions in black in the documents.
>     For the first time usage of the link, please accept the security certificate on your internet browser before proceeding. The website and the zip file are safe.
> * Updated all statistics and figures about the update size of the dataset (300 participants).
> * Introduced a new section that covers recommendations for future works to better design methods for A/H recognition (Sec.4)
> * Included new experiments using different ways of visual modality instead of cropped faces. This covers: full frame, and head-pose features.
> * Included analysis of annotator cues (Sec.H).
> * Included inter-annotator agreement analysis (Sec.2.1).
> * Included demographic statistics of train, validation, and test sets (Sec.G)
> * Included distribution of participants birth country (Sec.F)
> * The published BAH dataset is now shipped with the following additional information:
>     * Annotator identifier (ID)
>     * Subject meta-data including:
>         * Age of the participant.
>         * Age range of the participant.
>         * Birth country of the participant.
>         * Canada province where the participant lives.
>         * Ethnicity of the participant.
>         * Ethnicity simplified: a simplified version of the participant ethnicity.
>         * Sex of the participant.
>         * Is participant a student or not.
>         * Use in publications: can the videos and the data of the participant be used or not for publications.
>         * Use in challenges: can the videos and the data of the participant be used or not for challenges.
>     * Code to extract frames
>
> **W-1: Use video-based models.**
>
> **A**: Thank you for this comment. In our experiments, we used state-of-the-art multimodal and fusion techniques that won a recent affective computing challenge (Workshop and Competition on Affective & Behavior Analysis in-the-wild, 8th ABAW, 2025). Such a model allows dealing with multimodal input to perform frame level predictions. Each modality has its own backbone, followed by additional feature extraction layers. A temporal modeling is then used followed by fusion module. Indeed, it would be interesting and relevant  to explore more specialised video-based models. We are currently looking to integrate a recent video-based model, such as Video-FocalNet, to perform video-level prediction, before the end of the discussion period.

---

> > ### Author Response · Authors · 2025-11-21
> > **next response**
> >
> > **W-2/Q-2: How is A/H recognition different from standard action recognition?**
> >
> > **A**: Thank you for this critical question. Compared to activity recognition (where different parts of the body and different object can be involved), in A/H recognition always involves  the human body, and the differences of expression can be minimal and therefore difficult to infer. Compared to standard action recognition and video-based emotion recognition task, A/H recognition is even more challenging because the expressions are more subtle, vary considerably across different individuals, and are often associated with discord according to different modalities. Given these challenges, video-based A/H recognition must rely on multiple audio, visual and textual modalities, like cropped faces, vocal tone and transcripts.
> >
> > We included a new analysis of annotator cues in Sec. H. The results show that:
> >
> > 1- All cues (facial, language, body, and audio) participate at almost the same rate, showing conflict between 28.4% and 22.7% of the time; and
> >
> > 2- Their co-occurrence (facial, language, body, and audio) dominates with 54.1% followed by facial-body-language cues combination with 14.7%. Such critical analysis could be used a-priori for training and constraining a model to explicitly detect conflict before making the final A/H decision.
> >
> > In a new Section 4, we include several recommendations that we believe could help develop more powerful A/H recognition methods. The idea of using sentiment analysis at a first level is introduced. Each modality is trained separately for sentiment analysis using off-the-shelf models. The response of these models could be used in a second level to predict A/H. We also included modeling conflict using statistical priors from our cues analysis. Such a modular and interpretable framework is expected to be more efficient than generic multimodal models. Additionally, constraints on conflicts within and cross modalities can be easily introduced, something that is not directly feasible for standard models.
> >
> > **W-3: Dataset size is relatively small.**
> >
> > **A**: Thank you for this question. While our new dataset contains 8.26 hours (now: 10.60 hours, 300 participants, and 1427 videos), it is relatively large in the affective computing domain (see Table 8 in supplementary material provided in the initial submission). More importantly, the dataset is highly specialised. A/H cannot be reliably detected  in general videos like Youtube or other social media videos. It is more likely to surface in specific conversational contexts - in this case those that explore behaviour-related decisions, intentions, or conflicts. This is why we designed a structured 7-question elicitation protocol to reliably evoke and capture A/H in discussions about personal behaviours. To implement this, we built a custom online data-collection platform, recruited a pan-Canadian sample, obtained consent, and collected video responses under behavioural prompts; each recording was then reviewed and annotated by three trained experts. The result of more than 1 year of work, done by more than 12 people from machine learning and behavioral science teams, is data from 300 participants, 10.60 hours of context-rich videos containing 1.79 hours of A/H. It is a highly valuable dataset allowing for both machine learning training and behavioral science analyses.
> >
> > With the additional specialisation in A/H prediction, being participant (subject)-based, the availability of four supervision levels labeled by experts, being multimodal, and having a diverse population, makes the BAH dataset extremely valuable and useful for downstream tasks. Compared to recent affective computing datasets such as Aff-Wild2 (link: https://sites.google.com/view/dimitrioskollias/databases/aff-wild2) with 594 youtube videos, and C-EXPR-DB dataset with 400 videos, BAH dataset is larger (link: https://sites.google.com/view/dimitrioskollias/databases/c-expr-db). We believe that these features allow BAH dataset to be used to develop machine/deep learning models, and also in behavioral science studies.

---

> > > ### Author Response · Authors · 2025-11-21
> > > **next response**
> > >
> > > **Q-1: Limited performance over multimodal case.**
> > >
> > > **A**: Thank you for this comment. Indeed, multi-modal and contextual modeling have only slight benefit compared to single visual modality despite using state-of-the-art multimodal and fusion techniques that topped a recent affective computing challenge (8th Workshop and Competition on Affective & Behavior Analysis in-the-Wild, ABAW). In addition, combining the 3 modalities showed a decline in AP and slight drop in AVGF1, compared to the dual modality cases. Adding more modalities is often expected to improve accuracy in standard classification problems, assuming that the modalities reflect some diversity. However, it is not the case here. We believe that developing better and adapted fusion techniques will improve the performance. The role of fusion in our task is not simply to combine modalities. It should also assess affect in each modality, then determine if there are conflicting emotions, an important indicator of AH. The main goal of our experiments is to assess the performance of standard multimodal deep learning  techniques on the A/H recognition task, and highlight their limitations. Results of our experiments  serve as baselines for future works.

---

### Official Review · Reviewer_tpao · 2025-11-01

**Soundness:** 3
**Presentation:** 3
**Contribution:** 3
**Rating:** 6
**Confidence:** 3

**Summary:**

The paper proposes BAH, a dataset for recognizing ambivalence and hesitancy (A/H) in short webcam-style videos. The dataset consists of 1118 videos (~8 h total) from 224 participants across 9 Canadian provinces, each responding to 7 prompts. The dataset includes both frame-level and video-level A/H annotations. Baseline models are provided for visual, audio, and text modalities, as well as simple fusion and contextual vs non-contextual comparisons.

**Strengths:**

1. The first publicly available dataset specifically focused on ambivalence/hesitancy detection.
2. Clearly described data collection and annotation framework, incorporating both global- and frame-level labels

**Weaknesses:**

1. All subjects are from a single country (Canada), which may limit cultural and linguistic generalization of ambivalence expressions.
2. Contextual and multimodal (tri-modal) results are unexpectedly similar to non-contextual or single-modality baselines, suggesting limited exploitation of temporal or cross-modal dependencies.

**Questions:**

1. The contextual and multimodal (especially tri-modal) results are very close to those of single-modality models. How do you explain this? Why did tri-modal fusion underperform pairwise combinations? Are these differences statistically significant across runs?
2. Given that ambivalence is highly context-dependent, what temporal window or duration do you consider sufficient for a valid A/H judgment? How sensitive are the annotations or models to this choice?
3. How was ground truth established when annotators disagreed? Was it based on majority vote, adjudication by a lead annotator, or consensus discussion?
4. The codebook explicitly includes body language cues, yet the modeling pipeline omits body or pose features. Why didn’t you include these, and do you plan to in future work?
5. Can you provide inter-annotator reliability metrics (e.g. Cohen’s κ) for A/H labels? Also, how were temporal boundaries defined and aligned between annotators?

**Details Of Ethics Concerns:**

Dataset includes identifiable human subjects (faces, voices, transcripts) and is shared via public links with passwords. The paper does not specify whether data from participants who did not consent to public release were excluded.

---

> ### Author Response · Authors · 2025-11-21
> **Rebuttal**
>
> Thank you for the constructive review, and for acknowledging our contributions, including the novelty of the dataset, richness of annotation, and clarity of data collection and annotation framework. Similar positive feedback is shared by the other reviewers. We address each weakness and question mentioned below, and incorporate all feedback in the manuscript. “W” indicates weakness, “Q” indicates question, while “A” is our answer.
>
>
> Since the submission of this work, we have collected and annotated an additional 76 participants with a total of 309 videos. The final BAH dataset will have videos from 300 participants. We request permission from the Reviewer to update the dataset. We are working to reproduce all the results with this new version and update the manuscript before the end of the discussion period. In the current updated manuscript, we included the following changes (in red):
> * The link to the new BAH dataset with 300 participants: https://142.137.245.13/index.php/s/MyY2GyzBwjNXFLq (under folder: BAH_DB_300_ICLR26_REVIEW) , please use the same password provided in the initial submission to unzip the file:
> @ICRL_oSY5QhGTHH5ckAf3qKCF_2026_Brazil
>
>     Authors’ identity and relevant information have been redacted. Please do not select with mouse regions in black in the documents.
>      For the first time usage of the link, please accept the security certificate on your internet browser before proceeding. The website and the zip file are safe.
> * Updated all statistics and figures about the update size of the dataset (300 participants).
> * Introduced a new section that covers recommendations for future works to better design methods for A/H recognition (Sec.4)
> * Included new experiments using different ways of visual modality instead of cropped faces. This covers: full frame, and head-pose features.
> * Included analysis of annotator cues (Sec.H).
> * Included inter-annotator agreement analysis (Sec.2.1).
> * Included demographic statistics of train, validation, and test sets (Sec.G)
> * Included distribution of participants birth country (Sec.F)
> * The published BAH dataset is now shipped with the following additional information:
>     * Annotator identifier (ID)
>     * Subject meta-data including:
>         * Age of the participant.
>         * Age range of the participant.
>         * Birth country of the participant.
>         * Canada province where the participant lives.
>         * Ethnicity of the participant.
>         * Ethnicity simplified: a simplified version of the participant ethnicity.
>         * Sex of the participant.
>         * Is participant a student or not.
>         * Use in publications: can the videos and the data of the participant be used or not for publications.
>         * Use in challenges: can the videos and the data of the participant be used or not for challenges.
>     * Code to extract frames
>
> **W-1: Participants are limited to a single country.**
>
> **A**: Thank you for this important observation. While it is true that the data is sourced exclusively from Canadian participants, we would like to highlight that the sample was designed to be representative in terms of province and sex, two key demographic variables relevant to health behaviour research in Canada. Additionally, the sample includes individuals from a wide range of backgrounds, including different ethnicities and national origins, which reflects the cultural diversity of the Canadian population. This diversity, often lacking in many existing datasets, enhances the relevance and applicability of our findings within the Canadian context. We recognise that geographic limitation is a constraint, and we consider this dataset a meaningful starting point for understanding ambivalence and hesitancy in health-related behaviour. We agree that future work should expand to other countries and cultural settings to support broader generalisability, which is our plan. These points have been clarified in the revised manuscript (Sec. 2.1).

---

> > ### Author Response · Authors · 2025-11-21
> > **next reponse**
> >
> > **W-2/Q-1: Limited performance over multimodal case.**
> >
> > **A**: Thank you for this comment. Indeed, multi-modal and contextual modeling have only slight benefit compared to single visual modality despite using state-of-the-art multimodal and fusion techniques that topped a recent affective computing challenge (8th Workshop and Competition on Affective & Behavior Analysis in-the-Wild, ABAW). In addition, combining the 3 modalities showed a decline in AP and slight drop in AVGF1, compared to the dual modality cases. Adding more modalities is often expected to improve accuracy in standard classification problems, assuming that the modalities reflect some diversity. However, it is not the case here. We believe that developing better and adapted fusion techniques will improve the performance. The role of fusion in our task is not simply to combine modalities. It should also assess affect in each modality, then determine if there are conflicting emotions, an important indicator of AH. The main goal of our experiments is to assess the performance of standard multimodal deep learning  techniques on the A/H recognition task, and highlight their limitations. Results of our experiments  serve as baselines for future works.
> >
> > “Multiple Runs”: our results are reported using a single run. However, we will produce results for several runs  before the end of the discussion period by changing the random seed as our code is reproducible.
> >
> >
> > **Q-2: What is the impact of the size of the temporal window?**
> >
> > **A**: Thank you for this question. Indeed, the window size which reflects the duration of context is an important factor in a model's performance. In the initial manuscript version, we have provided in the supplementary materials an ablation about the window size (Sec.J.1, Ablation over the window length). From these results, a few seconds are necessary to capture A/H. For more certain predictions (better AP), small context is necessary (~3 secs). For better AVGF1, larger context is needed (~28 secs) in order to capture both classes. Note that the longest video has a length of 96 secs. This is also reflected in our statistical analysis of the dataset where average A/H segment duration is between 2-5 seconds (50-150 frames) (Sec. F). This has been a focus in our recommendations for future works (see Sec.4).
> >
> >
> > **Q-3-5: Clarification about the multi-annotation.**
> >
> > **A**: Thank you for this comment. Three annotators labeled the data. To ensure consistency and clarity, all annotators underwent structured training using a standardized protocol, supported by a cue list and annotation guidelines (codebook). Weekly meetings were held to discuss challenging or ambiguous cases, and collaborative review was encouraged during the training phase and when needed throughout annotation. This process helped align annotator interpretations and supported the development of shared understanding of the constructs. Only very few cases required this collective decision. The three annotators split the data participant-wise among themselves for fast annotation where each video was labeled by one annotator. Note that annotating a single video is a resource intensive process. The decision to divide the dataset among annotators was made to allow us to annotate a larger number of participants and ensure the dataset would be of sufficient size and diversity to support downstream machine learning applications. Our priority was to strike a balance between quality and scalability. We clarified this aspect in the main paper (see Sec. 2.1), and apologize for this confusion.
> >
> > To ensure the quality of the annotation, we perform a multi-annotation of a random subset of 10 videos to assess inter-annotator agreement. We considered Fleiss's kappa measure (REF1) At the global level, the score measure was 0.65 (substantial agreement) while at the frame level, it was 0.41 (moderate agreement). When considering only videos where the annotators agreed on the presence of A/H, the frame level score increased to 0.50. This evaluation is now included in the main paper (see Sec. 2.1).
> >
> >
> > [REF1]: Fleiss, J. L. (1971). Measuring nominal scale agreement among multiple raters. Psychological bulletin, 76(5), 378. https://doi.org/10.1037/h0031619

---

> > > ### Author Response · Authors · 2025-11-21
> > > **next response**
> > >
> > > **Q-4: What is the impact of using the full body as visual modality?**
> > >
> > > **A**: Thank you for this important comment. Indeed, body-language is an important cue for A/H detection. Our initial results were based only on cropped faces as they contain major cues. In addition, cropped faces are the dominant visual modality commonly used in affective computing. Following your comment, we conducted additional experiments where we considered full frame images for the visual modality instead of cropped faces in order to capture the full body. Additionally, a second experiment that considered extracting features from the face was conducted. In particular, we used head-pose estimation as an additional feature as you suggested. For head-pose, we used the pretrained model TokenHPE (REF2). We froze the feature extractor, and added on top of it an A/H classifier. Results showed that, while full frame (body) and cropped faces can yield competitive results, head-pose yielded poorer results. The full frame contained a lot of more global contextual information, such as the background, that may not be discriminant. Segmenting the body may lead to better performance. Additionally, combining both full body and head-pose could lead to better performance. Results are included in the main paper (Sec.3.3).
> > >
> > > [REF2]: Zhang, Cheng, et al. "Tokenhpe: Learning orientation tokens for efficient head pose estimation via transformers." CVPR. 2023. Code: https://github.com/zc2023/TokenHPE
> > >
> > > **Ethics concerts: Clarification about consent to public release.**
> > >
> > > **A**: Thank you for this important question. We took extreme care regarding ethics, privacy and consent for our data collection and sharing protocols. We confirm that all videos made publicly available in the BAH dataset contain only videos of participants that consented for their data to be part of a public dataset. This information also includes the meta-data being released. This is aligned with ethics of the dataset and the consent agreed on by the participants during the collection of their data.
> > > We further note that we obtained certification of ethical acceptability for research involving human subjects to collect this data from university human research ethics committees.
> > >
> > > This aspect has been further clarified in the main paper (Sec. 2.1)

---

> > > > ### Author Response · Authors · 2025-12-04
> > > > **update**
> > > >
> > > > Thank you again for your insightful comments. Due to space limitation and paper revision, the newly introduced results section Sec.3.3 (Visual Modality: Cropped Faces vs Full Frame vs Head-pose Features) has been moved to supplementary materials (Sec.J1).
> > > >
> > > > As for the question regarding multi-runs (Q-1), we conducted 5-runs with different seeds for studying the impact of using different variants of visual modalities (cropped faces, full frame, head-pose features). Performance and analysis did not change showing stability of our results. Additionally, using cropped faces showed higher stability (low variance) compared to the two other cases. Mean and standard deviation are reported for results in Sec.J1.

---

### Author Response · Authors · 2025-11-21
**General response to all reviewers**

We would like to express our sincere gratitude to all reviewers for their invaluable feedback throughout this review process. Their insightful questions and suggestions have significantly helped us clarify and strengthen our paper. We truly appreciate the positive recognition of our work shared by all four reviewers including the novelty of the task and multimodal dataset (tpao, RBNh, GdzA, pr3G), richness of modalities and annotation (tpao), demographic diversity (GdzA), well-designed data collection platform (tpao), and the valuable and comprehensive benchmarking (RBNh, GdzA, pr3G) that highlights the difficulty and challenges posed by the new task of video-based ambivalence/hesitancy (A/H) recognition.

Reviewers mainly requested clarifications on the difference between standard action recognition and A/H recognition in videos. Essentially, A/H recognition is more challenging as A/H are expressed in a subtle way, with high inter-person variability,  and is often associated with conflicting expressions across different modalities. State-of-the-art multimodal expression recognition models are ill equipped to accurately predict under such conflict. Additionally, reviewers requested additional interpretations of our results. Combining all modalities did not yield the best results compared to pairwise combinations. More specialized fusion techniques are needed to efficiently combine modalities for A/H recognition. Reviewers also asked for additional experiments with other visual modalities such body image, and head-pose features. Our results using full frame (with body) provides a wider context and provides competitive results compared to using cropped faces, yet head-pose features achieved lower performance. Moreover, reviewers requested further clarification on the annotation process. The three expert annotators divided the dataset to balance quantity and quality of annotation to construct such a large dataset for machine learning training. Additionally, our analysis of multi-annotator agreement over a random subset of videos showed a moderate to substantial agreement between global- and frame-level annotation.  All reviewer comments were answered below and integrated into the revised manuscript. We have also included new results with full frame and head-pose features.

Since the submission of our manuscript, we have collected and annotated 309 videos from 76 additional participants. The final BAH dataset will have a total of 1427 videos from 300 participants. This larger version of our dataset is available to reviewers. We request from all reviewers permission to update the dataset. The main updates to the manuscript for the dataset include an analysis of annotation cues; demographic statistics of train, validation, and test sets; the distribution of participants according to the country of birth; meta-data of participants with the dataset; an inter-annotator agreement analysis; and a new section for recommendations for designing future specialized methods for video-based AH recognition.

Finally, we would like to clarify that the purpose of our benchmarking work is mainly to provide a starting  baseline to initiate future works. Our results show also the limitations of standard and well-known models for multimodal expression recognition, fusion, and spatio-temporal. Researchers can compare to our results, and develop more specialized methods for video-based AH recognition.

---

### Author Response · Authors · 2025-12-04
**Final update to reviewers**

We thank all the reviewers again for their helpful comments. Please note that we have updated the abstract and introduction section to consider the reviewer  “GdzA” comments (W-1-2-3, Q-1-2-3) by focusing more on the newly introduced task. We have slightly updated the title as well for that reason. Due to space limitation, the newly introduced results section Sec.3.3 (Visual Modality: Cropped Faces vs Full Frame vs Head-pose Features) has been moved to supplementary materials (Sec.J1).

Please note that we have reproduced all results for frame- and video-level with the updated dataset version with 300 participants.
Results and discussions in the main paper and supplementary materials have been updated accordingly. Conclusions from our experiments remain the same where adapted fusion and temporal modules are critical to better leverage temporal and multimodal data for ambivalence/hesitancy recognition in videos.

We are in the process of completing the implementation of the Video-FocalNet [REF1] model for video-based prediction, an alternative of our used model, as suggested by reviewer “RBNh” in W-1. Running these experiments will take more time. Its results will be included in the camera-ready.

[REF1]: Wasim, Syed Talal, et al. "Video-focalnets: Spatio-temporal focal modulation for video action recognition." ICCV. 2023. Github: https://github.com/TalalWasim/Video-FocalNets

---

### Author Response · Authors · 2025-12-04
**Final message to Area Chair (1/3)**

Dear Area Chair,

Thank you for handling our submission, coordinating the insightful reviews, and for your service to the community. For your convenience, we briefly summarize our paper, its strengths, reviewers’ suggestions to improve our manuscript, comment about our benchmarking, and a list of the main changes in the manuscript.

**Summary of the paper**:

Our paper introduces video-based ambivalence/hesitancy (A/H) recognition, a challenging and critical task in digital health interventions. A/H, a closely related construct, is the primary reason for individuals to delay, avoid, or abandon health behaviour changes. This conflict sets a person in a state between positive and negative orientations, or between acceptance and refusal to do something, which often constitutes a barrier to initiating behaviour change and a trigger for discontinuing interventions or change efforts. A/H is a subtle and conflicting emotion manifested by a discord in affect between multiple modalities or within a modality, such as facial and vocal expressions, and body language.
While such affect cues are often detected and addressed during in-person interactions, there is currently no equivalent mechanism in digital health settings. Experts can be trained to identify A/H, integrating them into digital interventions but this is costly and less effective. This makes automatic A/H recognition critical in the personalization and effectiveness of digital behaviour change interventions as it provides a cost-effective alternative. However, there are currently no datasets available for the design of machine/deep learning models to recognize A/H.

This paper introduces the BAH dataset, a new multimodal and subject-based video dataset for affective computing and particularly for digital health interventions.  It is also large compared to existing affect datasets, with specialized 1,427 videos for a total duration of 10.60 hours, with 1.79 hours of A/H captured from a diverse set of 300 participants from across Canada answering predefined questions to elicit A/H. It is annotated by three behavioral science experts to include timestamp segments to indicate where A/H occurs, and provide frame- and video-level annotations with the A/H cues. Video transcripts, their timestamps, cropped and aligned faces, and participants' meta-data are also provided.

BAH is valuable for the  machine learning and behavior science communities, since it allows designing and testing new methods for real-world A/H recognition videos, in addition to providing new in-depth insight on A/H. Our initial benchmarking showed that existing baseline multimodal models yield limited performance. Specialized multimodal and spatio-temporal models are required to improve the performance of video-based A/H recognition, and for cost-effective digital health interventions.

**Paper’s strengths**:

We truly appreciate the positive recognition of our work shared by reviewers, including:
- The novelty of the task and multimodal dataset (tpao, RBNh, GdzA, pr3G),
- Richness of modalities and annotation (tpao), demographic diversity (GdzA),
- Well-designed data collection platform (tpao),
- The valuable and comprehensive benchmarking (RBNh, GdzA, pr3G) that highlights the difficulty and challenges posed by the new task of video-based ambivalence/hesitancy (A/H) recognition.

---

> ### Author Response · Authors · 2025-12-04
> **Final message to Area Chair (2/3)**
>
> **Reviewers’ suggestions to improve our manuscript**:
>
> - We clarified the difference between standard action recognition and A/H recognition in videos. Essentially, A/H recognition is more challenging because A/H is expressed in a subtle way, with high inter-person variability,  and is often associated with conflicting expressions across different modalities. State-of-the-art multimodal expression recognition models are ill equipped to accurately predict under such conflict. (Reviewers: “RBNh”, W-2/Q-2; “GdzA”, W-1-2-3/Q-1-2-3, “pr3G”, Q-4)
> - We improved the interpretation of our results. Combining all modalities did not yield the best results compared to pairwise combinations. More specialized fusion techniques are needed to efficiently combine modalities for A/H recognition. (Reviewers:  “tpao”, W-2/Q-1; “RBNh”, Q-1; “GdzA”, Q-4)
> - We added new experiments with other visual modalities such body image, and head-pose features. Our new results using full frame (with body) provides a wider context and provides competitive results compared to using cropped faces, yet integrating head-pose features achieved lower performance. (Reviewer: “tpao”, Q-4)
> - We provided further clarification on the annotation process. The three expert annotators divided the dataset to balance quantity and quality of annotation to construct such a large dataset for machine learning training. Additionally, our analysis of multi-annotator agreement over a random subset of videos showed a moderate to substantial agreement between global- and frame-level annotation. (Reviewers: “tpao”, Q-3-5; “pr3G”, W-3/Q-1)
>
> In our rebuttal below, we provided our detailed response to each concern raised by reviewers, and our modifications undertaken to improve the quality of the manuscript.

---

> > ### Author Response · Authors · 2025-12-04
> > **Final message to Area Chair (3/3)**
> >
> > **Benchmarking**:
> >
> > We would like to clarify that the purpose of our benchmarking work is mainly to provide initial results with  baseline models to initiate future research. Our results show also the limitations of standard and well-known baseline models for multimodal expression recognition, fusion, and spatio-temporal. Researchers can compare to our results, and develop more specialized methods for video-based AH recognition.
> >
> > **Manuscript changes (in red)**:
> > - They are mainly about clarifications suggested by reviewers.
> > - Revised abstract and introduction to cover  “GdzA” comments (W-1-2-3, Q-1-2-3) by focusing more on the newly introduced task. Following their same comment, we have also slightly adapted the paper title by adding the word ‘Digital’. The new title is “BAH Dataset for Ambivalence/Hesitancy Recognition in Videos for **Digital** Behavioural Change”. It highlights the relation of our work  to digital health applications.
> > - Updated all experimental results, statistics and figures about to account for the update size of the dataset (300 participants).
> > - Introduced a new section that covers recommendations for future works to better design methods for A/H recognition (Sec.4).
> > - Included new experiments using different ways of visual modality instead of cropped faces. This covers: full frame, and head-pose features.
> > - Included a new section: analysis of annotator cues (Sec.H).
> > - Included inter-annotator agreement analysis (Sec.2.1).
> > - Included a new section: demographic statistics of train, validation, and test sets (Sec.G).
> > - Included distribution of participants birth country (Sec.F)

---

### Meta-Review · Area_Chair_Ez7r · 2026-01-06

**Summary:**

The reviewers highlight several strengths of the work, noting that it introduces the first publicly available dataset specifically dedicated to ambivalence and hesitancy detection, addressing subtle and conflicting emotional states manifested across facial, vocal, linguistic, and bodily cues. The data collection and annotation process is clearly described and thoughtfully designed, incorporating both global and frame-level labels, which supports nuanced analysis. By framing ambivalence and hesitancy as complex, sustained affective states rather than basic emotions, the paper proposes a novel and meaningful affect recognition task that broadens the traditional emotion recognition paradigm. The dataset is of moderate scale, comprising videos from 224 participants across nine Canadian provinces, providing demographic and ethnic diversity that supports fairer modeling. Finally, the baseline experiments are well designed and comprehensive, covering frame- and video-level recognition in unimodal and multimodal settings, and are further strengthened by zero-shot and unsupervised domain adaptation experiments, enhancing the dataset’s utility and relevance for future research.

At the same time the reviewers raise concerns about the dataset’s scope, clarity, and scientific positioning: all subjects are drawn from a single country (Canada), limiting cultural and linguistic generalization, and the dataset is relatively small (8.26 hours), which may constrain its usefulness. Despite incorporating contextual and multimodal signals, experimental results are unexpectedly similar to single-modality baselines, suggesting limited exploitation of temporal or cross-modal information and an evaluation that relies only on image-based models rather than more appropriate video architectures. Annotation design is limited, relying on binary video-level labels without continuous affect dimensions or temporal localization, and the paper omits key reliability measures such as inter-annotator agreement and demographic statistics for dataset splits, weakening confidence in label quality, fairness, and overall research value.

**Reviewer Concerns:**

The reviews were fairly positive from the beginning with a sigle exception. There were two specific aspects that were well-defined in the rebuttal: the limited diversity of the subjects and the unexpectedly similar experimental results to single-modality baselines despite incorporating contextual and multimodal signals. The subject and the dataset are valuable even if the subject is a bit of an outlier. Still, all the reviewers appreciate its importance and the contribution of affective computing is quite clear. There are still left the limited scope and size of the dataset but the latter is a moving target and the authors already indicated that the size has been increased.

**Reviewer Scores:**

I think the discussion would have been beneficial for clarifying some of the points raised by the reviewers but I think the authors have correctly answered the concerns and the comments were not critical from the beginning. The only concern I still have is the limited scope of the dataset but this should not be a critical point.

---

### Decision · Program_Chairs · 2026-01-26

Accept (Poster)